# Latent Radiance Fields with 3D-aware 2D Representations

**Chaoyi Zhou**[*]**, Xi Liu,**[*] **Feng Luo, Siyu Huang**[†]
Visual Computing Division
School of Computing
Clemson University
{chaoyiz,xi9,luofeng,siyuh}@clemson.edu

Groundtruth  3DGS on Latent Space  Latent Radiance Fields (ours)

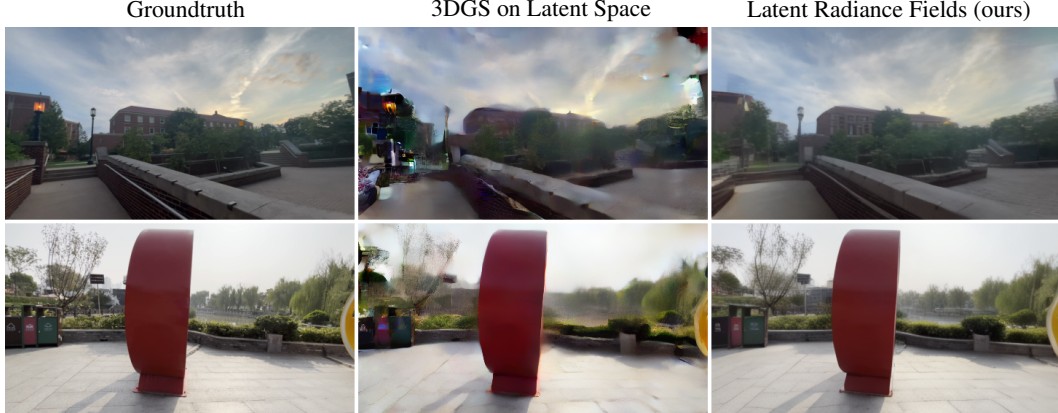

Figure 1: This work novelly enables the radiance field representations on the latent space of VAE, achieving photorealistic 3D reconstruction performance on unbounded outdoor scenes.

## ABSTRACT

Latent 3D reconstruction has shown great promise in empowering 3D semantic understanding and 3D generation by distilling 2D features into the 3D space. However, existing approaches struggle with the domain gap between 2D feature space and 3D representations, resulting in degraded rendering performance. To address this challenge, we propose a novel framework that integrates 3D awareness into the 2D latent space. The framework consists of three stages: (1) a correspondence-aware autoencoding method that enhances the 3D consistency of 2D latent representations, (2) a latent radiance field (LRF) that lifts these 3D-aware 2D representations into 3D space, and (3) a VAE-Radiance Field (VAE-RF) alignment strategy that improves image decoding from the rendered 2D representations. Extensive experiments demonstrate that our method outperforms the state-of-the-art latent 3D reconstruction approaches in terms of synthesis performance and cross-dataset generalizability across diverse indoor and outdoor scenes. To our knowledge, this is the first work showing the radiance field representations constructed from 2D latent representations can yield photorealistic 3D reconstruction performance. The project page is latent-radiance-field.github.io.

## 1 INTRODUCTION

Recently, significant advancement in radiance field representation, such as Neural Radiance Fields (NeRF) (Mildenhall et al., 2020) and 3D Gaussian Splatting (3DGS) (Kerbl et al., 2023), have been made for fast and high-quality 3D reconstruction and novel view synthesis (NVS). As demonstrated

---

[*]*Equal contribution*
[†]*Corresponding author: Siyu Huang*

by Stable Diffusion Models (Rombach et al., 2021), optimizing in the 2D latent space instead of the image space can significantly boost generation efficiency. Meanwhile, a 3D-consistent latent space and photorealistic decoding capability can benefit many tasks such as text-to-3D generation, latent NVS, few-shot NVS, efficient NVS, 3D latent diffusion model, and 3D semantic understanding. To empower 3D semantic understanding, researchers have explored latent 3D reconstruction methods, such as Feature 3DGS (Zhou et al., 2024a), to distill 2D semantic features into 3D space for novel view semantic segmentation. However, there are significant domain gaps between the 2D feature space and 3D representations, arising from the lack of consistent 3D spatial structure information, which hinders the direct feeding of 2D features into the 3D representations. The 2D feature extractors cannot effectively perceive the 3D structures behind the inputs images since the training images are presented to the network in an unstructured way and the training objective does not include 3D consistency. Therefore, the loss of 3D awareness is inevitable.

Few previous work attempt to bridge the gap between the 2D features and 3D representations. By focusing on better 3D scene understanding, Feature 3DGS (Zhou et al., 2024a) proposes to distill a feature field from 2D semantic features by leveraging the view-independent approach while it cannot model the view-dependent visual properties. Another line of work improves the latent field in the context of 3D generation task. Latent-NeRF (Metzer et al., 2022) and ED-NeRF (Park et al., 2023) includes an additional per-scene refinement layer to enhance the latent rendering quality, while introducing more computation cost and exhibiting the generalizabilty.

To bridge the gap between the 2D latent space and 3D representations, we observe two main challenges: Firstly, the massive view-dependent high-frequency noise in the 2D latent space causes the inconsistent geometry and unstable optimization. Moreover, data distribution shift by applying RGB-based NVS methods to latent features also prohibits the photorealistic rendering. To tackle with these two issues, our key insight is to embed 3D awareness into the latent space, while maximumly preserving the representation ability of autoencoders without introducing any additional layers. We propose a novel framework that builds a latent radiance field (LRF) based on the 3D-aware 2D representations. Specifically, it consists of three stages. Firstly, we introduce a correspondence-aware autoencoding method to improve the 3D awareness of the VAE's latent space, making the 2D representations follow the geometry consistency. Then, we build the LRF to represent 3D scenes from the 3D-aware 2D representations, lifting the 3D-aware 2D representations into the 3D space. Finally, we introduce a VAE-Radiance Field(VAE-RF) alignment method to further mitigate the data distribution shift caused by NVS and boost the performance of image decoding from the rendered 2D representations. In together, the created 3D-aware latent space and LRFs can be smoothly injected into existing NVS or 3D generation pipelines without further fine-tuning, achieving high-quality and photorealistic synthesis results.

To the best of our knowledge, this is the first work demonstrating that radiance field representations constructed in the latent space, with the injection of 3D awareness, can achieve photorealistic 3D reconstruction performance across various settings including indoor and unbounded outdoor scenes. Extensive NVS, 3D generation, and few-shot novel view synthesis experiments show that our method outperforms existing methods with respect to its high-quality synthesis and cross-dataset generalizability, as shown in Fig. 1 and the following sections. In summary, main contributions of this work include:

- We introduce a novel framework to integrate 3D awareness into 2D representation learning, including a correspondence-aware autoencoding method and a VAE-Radiance (VAE-RF) field alignment to enable high-quality 3D reconstruction in latent space.

- We propose the latent radiance field (LRF) to effectively elevate the 3D-aware 2D representations into 3D latent fields. It represents the first step towards constructing radiance field representations directly in the latent space for 3D reconstruction tasks.

- We conduct extensive experiments to show that our method achieves superior fidelity and cross-dataset generalizability across NVS, few-shot NVS, and 3D generation tasks.

## 2 RELATED WORK

**Injecting 3D priors into 2D representations.** While many existing works focus on incorporating 2D features into 3D representations, which improves performance in downstream tasks such as

scene understanding (Zhi et al., 2021; Ha & Song, 2022; Qin et al., 2023; Shi et al., 2023; Zhou et al., 2024a; Cen et al., 2023; Gu et al., 2024; Guo et al., 2024), less attention has been paid to the opposite direction: leveraging 3D knowledge to enhance 2D features, which benefit challenging tasks that require 3D understanding while the perceived information is limited such as monocular depth estimation (Stan et al., 2023; Bhat et al., 2023; Piccinelli et al., 2024; Chatterjee et al., 2024; Moon et al., 2023) and semantic segmentation (Wang et al., 2023; Sun et al., 2024). Studies such as (Bachmann et al., 2022; Zhou et al., 2024a) utilize 3D priors from multi-view and geometric information to improve the Masked Autoencoders (He et al., 2021), achieving better performance on downstream tasks of segmentation and detection. However, directly injecting the geometry constraints into the pre-trained feature extractors is harmful for the self-supervised 2D representation and heavily relying on pre-trained feature extractors poses potential limitations for performance and requires significant computational resources. In contrast, our method does not require any additional per-scene refinement module, serving as an efficient and generalizable approach for injecting 3D priors into 2D representations.

**Radiance field representations on images and features.** Neural Radiance Fields (NeRF) (Mildenhall et al., 2020) and 3D Gaussian Splatting (3DGS) (Mildenhall et al., 2020) are benchmark radiance field representation methods for the NVS task. NeRF represents 3D scenes and renders photorealistic novel views based on the representation capacity of neural networks. 3DGS employs a set of 3D Gaussian primitives to represent 3D scenes, and a fast differentiable rasterizer to enable more efficient rendering while keeping the photorealism of novel views. However, the distillation of the 2D features into the 3D representations remains challenging, mainly due to the significant geometric inconsistency in the feature maps caused by massive high-frequency information. Therefore, some recent literature (Zhou et al., 2024a; Kobayashi et al., 2022; Siddiqui et al., 2023; Fan et al., 2022; Kerr et al., 2023) propose alternative solutions by leveraging the geometry information from the RGB space to help the 3D reconstructions of 2D features. Fit3D (Yue et al., 2024) builds a huge amount of 3D representation dataset as the superivsion for the pre-trained feature extractor fine-tuning; however, without considering the compatibility of the 3D representation and 2D feature space, they also require a customized decoder to ensure the performance in the downstream tasks. All the methods mentioned above all rely on the per-scene optimization with additional modules, while our method bridging the gap between 2D feature space and 3D representation with an efficient correspondence-aware method.

**Text-to-3D generation with 2D priors.** Despite the impressive 3D generation capabilities demonstrated by many existing 2D generative prior-guided works (Tang et al., 2023; Poole et al., 2022; Wu et al., 2023; Zhou et al., 2024b; Jain et al., 2022; Michel et al., 2021), performing back-propagation of the Score Distillation Sampling (SDS) loss (Poole et al., 2022) on images is computationally intensive and time-consuming. Latent diffusion models (LDMs) offer more efficient solutions by operating in the latent space. However, the vastly different distribution of the latent space means that directly utilizing the latent representations for NVS leads to degraded rendering performance. To our knowledge, only a few works attempt to overcome this challenging task. Latent-NeRF (Metzer et al., 2022) employs a per-scene refinement layer to map the rendered latent to RGB space as an additional constraint for training the NeRF representations. ED-NeRF (Park et al., 2023) introduces a more complex refinement module by initializing from a set of specific layers in a Variational Autoencoder (VAE). Although these per-scene refinement modules effectively mitigate the artifacts in the rendering results, they require resource-consuming optimization for each scene, and lack generalization ability to novel views or scenes. Moreover, the smoothness introduced by the neural networks hinders the reconstruction of high-frequency signals on the 2D features. On the contrary, our method requires no additional efforts for lifting the 2D features to the 3D radiance field representations, such that it can be injected into any existing NVS or text-to-3D frameworks smoothly and efficiently.

## 3 PRELIMINARIES

**Variational autoencoder.** A variational autoencoder (VAE) (Kingma, 2013) is a generative model that represents high-dimensional data distributions in a lower-dimensional latent space. The encoder maps the input data $\mathbf{x}$ to a latent variable $\mathbf{z}$ by estimating the parameters of a posterior distribution $q_\phi(\mathbf{z}|\mathbf{x})$. The posterior is typically assumed to follow the Gaussian distribution, parameterized by a mean $\mu_\phi(\mathbf{x})$ and a variance $\sigma_\phi(\mathbf{x})$. The latent variable $\mathbf{z}$ is sampled from this posterior distribution,

i.e., $\mathbf{z} \sim q_\phi(\mathbf{z}|\mathbf{x}) = \mathcal{N}(\mathbf{z}; \mu_\phi(\mathbf{x}), \sigma_\phi(\mathbf{x})^2)$. The decoder reconstructs the input $\mathbf{x}$ by mapping $\mathbf{z}$ back to the data space through the likelihood $p_\theta(\mathbf{x}|\mathbf{z})$. The learning objective of is:

$$\mathcal{L}_{\text{VAE}}(\theta, \phi; \mathbf{X}) = \mathbb{E}_{q_\phi(\mathbf{Z}|\mathbf{X})}[\log p_\theta(\mathbf{X}|\mathbf{Z})] - \text{KL}(q_\phi(\mathbf{Z}|\mathbf{X})\|p(\mathbf{Z})). \tag{1}$$

**3D Gaussian Splatting.** 3DGS (Kerbl et al., 2023) is an efficient NVS framework that uses a set of 3D Gaussian primitives to represent a scene explicitly. Each Gaussian primitive has a position vector $\boldsymbol{\mu} \in \mathbb{R}^3$, a 3D covariance matrix $\boldsymbol{\Sigma} \in \mathbb{R}^{3\times3}$, an opacity $\alpha \in \mathbb{R}$, and a spherical harmonics (SH) coefficient $\boldsymbol{c} \in \mathbb{R}^k$ (Ramamoorthi & Hanrahan, 2001) representing the view dependent colors.

$$G(x) = e^{-\frac{1}{2}(x-\boldsymbol{\mu})^T \boldsymbol{\Sigma}^{-1}(x-\boldsymbol{\mu})}, \tag{2}$$

where $\Sigma = RSS^T R^T$, $S$ denotes the scaling matrix and $R$ is the rotation matrix. Then, rasterization (Zwicker et al., 2001) can transform the 3D Gaussian spheres to the 2D camera plane to calculate the 2D covariance matrix in the camera space as

$$\Sigma' = JW\Sigma W^T J^T, \tag{3}$$

where $W$ is the perspective transformation matrix and $J$ is Jacobin of the projection matrix. For every pixel, the Gaussians are traversed in depth order from the image plane, and their pixel colors $c_i$ are combined through alpha compositing, forming pixel color $C$ as

$$C = \sum_{i \in N} c_i \alpha_i \prod_{j=1}^{i-1} (1 - \alpha_j). \tag{4}$$

## 4 METHOD

In this work, we propose a method to achieve 3D-aware 2D representations and enable 3D reconstruction in the latent space. We base our method on the widely used Variational Autoencoder (VAE) from Latent Diffusion models (Metzer et al., 2022). To enhance the 3D awareness of both encoder and decoder of the VAE, we present a three-stage pipeline as illustrated in Fig. 2. The first stage focuses on improving the 3D awaresness of the VAE's encoder through a novel correspondence-aware constraint on the latent space, making the 2D representations follow the geometry consistency (Sec. 4.1); The second stage builds a latent radiance field (LRF) to represent 3D scenes from the 3D-aware 2D representations (Sec. 4.2); The third stage further introduces a VAE-Radiance Field (VAE-RF) alignment method to boost the reconstruction performance (Sec. 4.3). In together, our LRF enables 3D reconstruction on the 2D latent space instead of the image space. It can render high-quality and photorealistic novel views, even for the unbounded scenes (Sec. 5). More details of our method are discussed in the following sections.

### 4.1 CORRESPONDECE-AWARE AUTOENCODING

The first stage of our method is incorporating the geometry-awareness into the autoencoding process. Given $K$ muilt-view images $\mathcal{I} = \{\boldsymbol{I}_i\}_{i=1}^K, \left(\boldsymbol{I}_i \in \mathbb{R}^{H\times W\times 3}\right)$, the VAE encoder extracts 2D representations $\mathcal{Z} = \{\boldsymbol{Z}_i\}_{i=1}^K, \left(\boldsymbol{Z}_i \in \mathbb{R}^{H'\times W'\times 4}\right)$ in a low-dimensional latent space while the semantic information can be preserved effectively. However, as shown in Fig. 4, most of existing NVS frameworks fail to reconstruct the photo-realistic images from the rendered latents. It is mainly because the VAE encoding process significantly damages the multi-view consistency within the original image space, since the latent space presents massive high-frequency noises to compress the original RGB space into a compact latent space (see Fig. 3). This brings severe challenges for reconstructing the 2D latent representations in the 3D space.

**Correspondence consistency on the latent space.** To address the above issue and enable effective latent 3D reconstruction, we are inspired by the multi-view correspondence consistency which serves as the foundation principle for modeling the natural 3D world. Specifically, points $\boldsymbol{x}_i \in \mathbb{R}^2$ in image $\boldsymbol{I}_i$ and points $\boldsymbol{x}_j \in \mathbb{R}^2$ in another image $\boldsymbol{I}_j$ are considered correspondences if they are connected by the fundamental matrix $\boldsymbol{F}_{ij} \in \mathbb{R}^{3\times3}$, satisfying the multi-view geometry constraint (Schönberger & Frahm, 2016):

$$\boldsymbol{x}_j^\top \boldsymbol{F}_{ij} \boldsymbol{x}_i = 0. \tag{5}$$

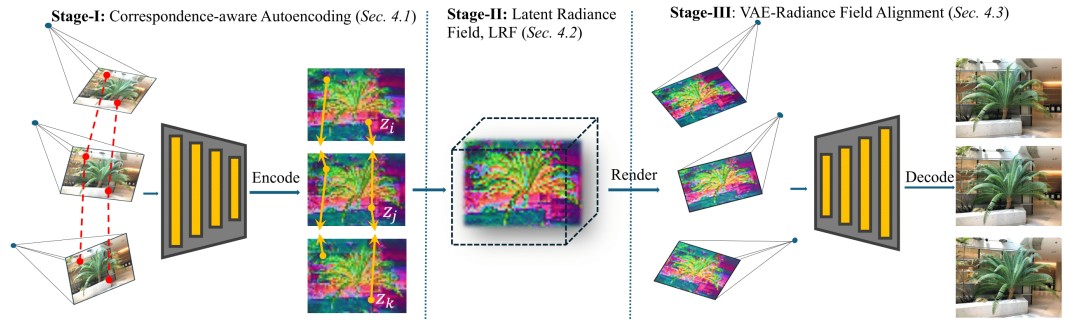

Figure 2: An illustration of our pipeline for creating a latent radiance field in conjunction with 3D-aware 2D representation fine-tuning. Firstly in Stage-I, we inject 3D awareness into the VAE's encoder through applying a novel correspondence consistency constraint on the latent space, making the 2D representations follow the geometry consistency. Then in Stage-II, we create the latent radiance field (LRF) to represent 3D scenes based on the 3D-aware 2D representations. Finally in Stage-III, we introduce a VAE-Radiance Field alignment method to enhance the performance of image decoding from the rendered latent space.

Eq. 5 tells that a pair of correspondence points on the image space should be close to each other, so that the consistent geometry can be ensured during the optimization in the 3D space; otherwise, the artifacts and redundant geometry representation due to the local optimal will damage the quality of the 3D reconstruction and novel view synthesize. Motivated by this, we propose an computationally efficient strategy that incorporates the correspondence consistency into the autoencoder training. Specifically, a set of multi-view images $\mathcal{I} = \{\boldsymbol{I}_i\}_{i=1}^K, \left(\boldsymbol{I}_i \in \mathbb{R}^{H \times W \times 3}\right)$ are fed into the autoencoder to extract the latent representations $\mathcal{Z} = \{\boldsymbol{Z}_i\}_{i=1}^K, \left(\boldsymbol{Z}_i \in \mathbb{R}^{H' \times W' \times 4}\right)$, and the correspondence consistency loss on the latent space is computed by

$$\mathcal{L}_{\text{corres}} = \sum_{i=1}^K \sum_{j \in \mathcal{K}(i)} \lambda_{ij} \left\| \boldsymbol{z}_i - \boldsymbol{z}_j \right\|_1 . \tag{6}$$

where $\boldsymbol{z}_i$ refers to the the latent pixel in the $\boldsymbol{Z}_i$ and $\boldsymbol{z}_i$ is the corresponding latent pixel in the neighbouring latent $\boldsymbol{Z}_j$. $\mathcal{L}_{\text{corres}}$ ensures that the encoded features follow the correspondence consistency derived from the multi-view images, where $\lambda_{ij}$ is the weight based on the average pose error (APE) calculated from the Frobenius norm between the two camera poses of images $\boldsymbol{I}_i$ and $\boldsymbol{I}_j$ to weight the accurate pose distance to represent the view-dependant latent codes. The detail of calculating $\lambda_{ij}$ can be found in Appendix A.1 By injecting the latent correspondence consistency into the standard VAE training, our VAE training objective is:

$$\mathcal{L}_{\text{StageI}} = \mathcal{L}_{\text{VAE}} + \lambda_1 \mathcal{L}_{\text{corres}} + \lambda_2 \mathcal{L}_{\text{reg}} . \tag{7}$$

$\mathcal{L}_{\text{VAE}}$ is original VAE traning objective for VAE in Eq. 1. $\mathcal{L}_{\text{reg}} = -\text{KL}\left(q(\boldsymbol{Z}|\boldsymbol{X}) \parallel q_{\text{original}}(\boldsymbol{Z}|\boldsymbol{X})\right)$ enforces the fine-tuned 2D representations being close to those of the pre-trained VAE, preserving the representation capability of the finet-tuned autoencoder. This new learning objective ensures that the compact latent space of VAE preserves the multi-view geometric consistency, such that it is compatible with existing NVS frameworks such as 3DGS.

**Insight into latent correspondence consistency.** The maximum degree of the spherical harmonics is always set as 3 in NVS methods for the efficiency and robustness in the modeling the view-dependant information. To be more specific, the lower degree terms is aim to mostly capture low-frequency information such as albedo for the scene while the higher degrees are tended to model the high-frequency, view dependent information such as the lightning. For the latent space, the latent code can be considered as the combination of the base value and high frequency noise. Due to such a compact representation, the amount of the noise can be greatly increase compared to the RGB space, creating more difficulties for the SH coefficients to model the information from different views. When maximum degree is fixed, it is easier for SH coefficients to reach the global optimal instead of locally over-fitting. Fortunately, with our $\mathcal{L}_{\text{corres}}$, the high frequency noise can be

| Image | VAE latent | Finetuned latent | VAE latent FFT | Finetuned latent FFT |
|---|---|---|---|---|

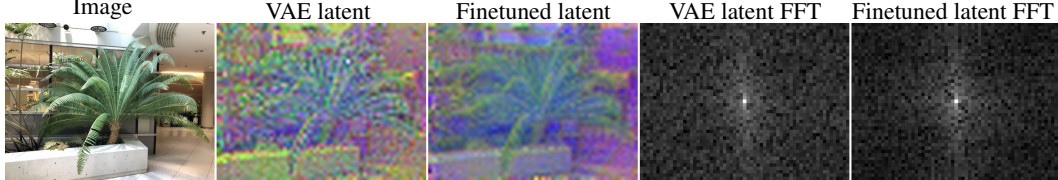

Figure 3: A visualization of latent spaces of original and our fine-tuned VAEs. Our method ensures an accurate geometry in the latent space while removing a certain amount of high-frequency noises.

effectively removed while the high-quality image generative ability can still be preserved, leading to a more stable process of the optimization and consistent geometry representation. Fig. 3 shows that the correspondence-aware encoding can significantly remove the high frequency noises in the 2D latent space and the visualization of applying Fast Fourier transform also showing less high-frequency noise in latent space achieved by our encoder, resulting an effective approach to lifting the 2D features into the 3D latent fields.

## 4.2 LATENT RADIANCE FIELD

Based on the 3D-aware 2D representation fine-tuning discussed in Sec. 4.1, we create 3D representations directly in the 2D latent space of VAE, namely the latent radiance field (LRF). We take 3DGS (Kerbl et al., 2023) as an example of radiance field representations to discuss our LRF.

By following 3DGS, a set of latent 3D Gaussians is formulated as

$$\mathcal{G} = \{(\boldsymbol{\mu}, \mathbf{s}, \mathbf{R}, \alpha, \mathbf{SH}_f)_j)\}_{1 \leq j \leq M}, \tag{8}$$

where $\boldsymbol{\mu} \in \mathbb{R}^3$ is the 3D mean of the Gaussian, $\mathbf{S} = \text{diag}(\mathbf{s}) \in \mathbb{R}^{3 \times 3}$ is the Gaussian scale, $\mathbf{R} \in \mathbb{R}^{3 \times 3}$ its orientation, $\alpha \in \mathbb{R}$ a per-Gaussian opacity, and $\mathbf{SH}_f$ models the view-dependant latent in the 3D latent space. By following the differentiable rasterization process of 3DGS, we rasterize the 2D latent representations using point-based $\alpha$-blending as follows:

$$\mathbf{Z} = \sum_{i \in \mathcal{N}} \mathbf{z}_i \alpha_i \prod_{j=1}^{i-1} (1 - \alpha_i), \tag{9}$$

where $\mathcal{N}$ is a set of ordered Gaussians overlapping the pixel, $\mathbf{z}_i \in \mathbb{R}^{dim}$ is the view-dependent latent code of each Gaussian, where $\mathbf{dim}$ is the number of the latent dimension of the feature. and $\alpha_i$ is given by evaluating a 2D Gaussian with covariance $\boldsymbol{\Sigma}$ multiplied with a learned per-point opacity. Let $\mathcal{I} = \{\boldsymbol{I}_i\}_{i=1}^K, (\boldsymbol{I}_i \in \mathbb{R}^{H \times W \times 3})$ be a set of multi-view images of a scene with corresponding camera parameters. Let $\mathcal{Z} = \{\boldsymbol{Z}_i\}_{i=1}^K, (\boldsymbol{Z}_i \in \mathbb{R}^{H \times W \times 3})$ be a corresponding set of latents from the VAE encoder. The rasterization function $r$ renders a set of latent Gaussians into a 2D latent representation according to the camera pose $\mathbf{P}_i$. Then, we optimize the latent Gaussian parameters, to optimally represent latent $\mathcal{Z}$:

$$\hat{\mathcal{G}} = \underset{\{(\boldsymbol{\mu}, \mathbf{s}, \mathbf{R}, \alpha, \mathbf{SH}_f\}}{\arg\min} \sum_{i=1}^N \mathcal{L}^f(r(\mathcal{G}, \mathbf{P}_i), \mathbf{Z}_i), \tag{10}$$

where $\mathcal{L}^f$ is a pixel-wise $l_1$ loss combined with a D-SSIM term. Notably, we do not need to impose additional geometric consistency constraints introduced by previous literature (Yue et al., 2024; Kobayashi et al., 2022; Zhou et al., 2024a), as our correspondence-aware autoencoder fine-tuning ensures geometrically consistent 2D representations in the 3D space. Therefore, our LRF reconstructs the 2D latent representations as a radiance field representation directly, and enables an efficient rendering of the 2D latent representations for novel views.

## 4.3 VAE-RADIANCE FIELD ALIGNMENT

Although the correspoondence-aware autoencoding introduced in Sec. 4.1 improves the 3D consistency of VAE latent space, the LRF distribution $\boldsymbol{p}(z_{\text{NVS}})$ are still shifted from the VAE latent

Table 1: Our method outperforms the image and latent space NVS baselines on most settings and metrics, from object-level to unbounded outdoor scenes. Latent-NeRF* denotes we adapt it to NVS.

| Dataset | Metric | Image Space | | Latent Space | | | |
| | | 3DGS/8 | Mip-Splatting/8 | 3DGS-VAE | Latent-NeRF* | Feature-GS | 3DGS-LRF (Ours) |
|---|---|---|---|---|---|---|---|
| **MVImgNet** | PSNR ↑ | 16.93 | 24.89 | 25.04 | 18.50 | 21.09 | 26.26 |
| | SSIM ↑ | 0.561 | 0.799 | 0.824 | 0.709 | 0.772 | 0.863 |
| | LPIPS ↓ | 0.466 | 0.328 | 0.250 | 0.403 | 0.372 | 0.178 |
| **NeRF-LLFF** | PSNR ↑ | 9.98 | 19.68 | 19.07 | 18.31 | 16.48 | 20.00 |
| | SSIM ↑ | 0.110 | 0.484 | 0.493 | 0.457 | 0.415 | 0.541 |
| | LPIPS ↓ | 0.631 | 0.513 | 0.364 | 0.387 | 0.539 | 0.289 |
| **DL3DV-10K** | PSNR ↑ | 14.03 | 21.81 | 20.57 | 18.16 | 16.60 | 22.45 |
| | SSIM ↑ | 0.352 | 0.609 | 0.595 | 0.530 | 0.449 | 0.667 |
| | LPIPS ↓ | 0.541 | 0.451 | 0.346 | 0.432 | 0.602 | 0.197 |
| **Mip-NeRF360** | PSNR ↑ | 14.79 | 22.38 | 19.44 | 15.93 | 17.13 | 20.83 |
| | SSIM ↑ | 0.273 | 0.502 | 0.404 | 0.312 | 0.337 | 0.469 |
| | LPIPS ↓ | 0.586 | 0.521 | 0.432 | 0.537 | 0.642 | 0.328 |

distribution $p(z_{\text{VAE}})$ due to the non-linearity in neural rendering, resulting in performance decrease when we decode LRF rendering results back to images through the VAE decoder.

We further propose to fine-tune the VAE decoder under the radiance field guidance to address this issue. With the LRF built in Sec. 4.2, we can reconstruct LRFs from a large amount of scenes to generate a latent-image paired dataset. This dataset consists of the 2D latent representations $\mathcal{Z} = \{\boldsymbol{Z}_i\}_{i=1}^K, \left(\boldsymbol{Z}_i \in \mathbb{R}^{H' \times W' \times 4}\right)$ rendered by LRFs and the corresponding ground truth images $\mathcal{I} = \{\boldsymbol{I}_i\}_{i=1}^K, \left(\boldsymbol{I}_i \in \mathbb{R}^{H \times W \times 3}\right)$. Notably, we also include the training views of LRFs in this dataset, since a key feature of existing NVS methods is to overfit the training views. The training objective of our VAE-RF alignment decoder fine-tuning is:

$$\mathcal{L}_{\text{StageIII}} = \lambda_{\text{train}} \|D(Z_{\text{train}}) - I_{\text{train}}\|_1 + \lambda_{\text{novel}} \|D(Z_{\text{novel}}) - I_{\text{novel}}\|_1, \qquad (11)$$

where $D(\cdot)$ is the decoder, $Z_{\text{train}}$ and $Z_{\text{novel}}$ are the latent codes of the training views and novel views, respectively. $I$ refer to the corresponding ground truth images. $\lambda_{\text{novel}}$ and $\lambda_{\text{novel}}$ are the weighting coefficient that balances the contributions of the training and novel views. Both of the weights are set to $0.5$ to ensure that the decoder learns not only to decode effectively from the training views but also to generalize and perform well on the novel views. Eq. 11 effectively minimizes the distribution mismatch between the VAE latent space and the LRF rendering space. After decoder fine-tuning, high-quality images can be reconstructed from the LRF rendering of either training or novel views. The fine-tuned autoencoder enhances 3D reconstruction and generation by providing a geometry-aware 2D latent space as well as a radiance field-compatible autoencoder.

# 5 EXPERIMENTS

## 5.1 LATENT 3D RECONSTRUCTION

We first evaluate LRF on four real-world datasets, including MVImgNet (Yu et al., 2023), NeRF-LLFF (Mildenhall et al., 2019), MipNeRF360 (Barron et al., 2022), and DL3DV-10K Ling et al. (2024), to demonstrate the effectiveness of our approach for latent 3D reconstruction. Among these datasets, DL3DV serves as an in-distribution dataset, where the training set is used for model training, and the test set is used for evaluation. In contrast, MVImgNet, LLFF, and Mip-NeRF360 are out-of-distribution datasets, as they have never been used in the training process. We follow the standard train and test split in 3DGS and Mip-Splatting(Kerbl et al., 2023; Yu et al., 2024).

Fig. 4 shows that our method significantly improves the capability of the 2D latent representations for 3D reconstruction task. Our approach mitigates the artifacts such as ghosting, color distortion, blurring, and texture warping caused by 3D inconsistency. While the latent and image space approaches share the same input resolution, our rendering results present clearer visual details, richer textures, and more high-frequency information.

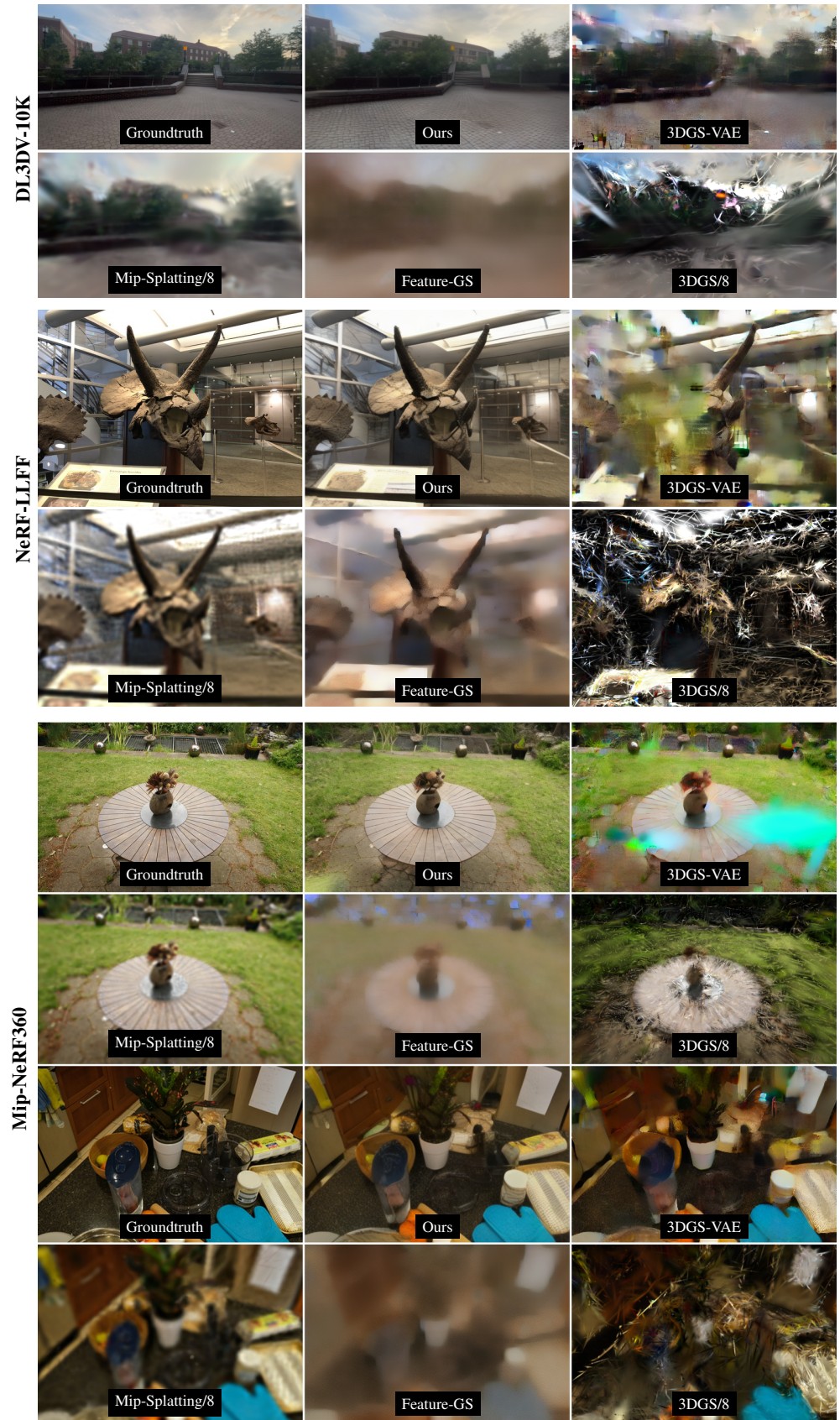

Figure 4: A visual comparison of rendering results. Our method can not only render high-quality images for in-distribution dataset (DL3DV-10K), but also shows great generalization ability across different datasets.

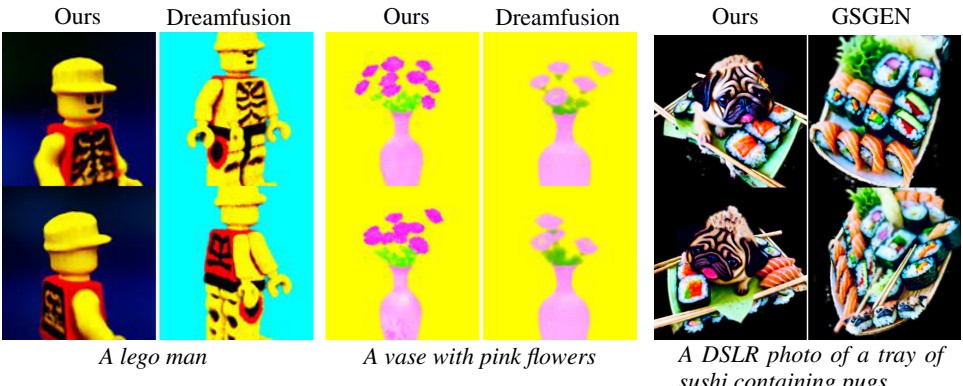

| Ours | Dreamfusion | Ours | Dreamfusion | Ours | GSGEN |
|------|-------------|------|-------------|------|-------|

*A lego man*      *A vase with pink flowers*      *A DSLR photo of a tray of sushi containing pugs*

Figure 5: Visual comparison of different text-to-3D generation methods. Our model enables the generation of more view-consistent results.

As shown in Table 1, LRF achieves the state-of-the-art performance across all datasets in terms of metrics of PSNR, SSIM, and LPIPS. These results underscore the effectiveness of our approach in fine-tuning latent space representations to support novel view synthesis. This demonstrates that our fine-tuning approach not only effectively reduces the geometry information loss caused by 3D-inconsistent 2D representations but also preserves

Table 2: A comparison of different methods on LLFF dataset using 3 views.

| Method | PSNR ↑ | SSIM ↑ | LPIPS ↓ |
|--------|--------|--------|---------|
| 3DGS-VAE | 13.06 | 0.283 | 0.570 |
| 3DGS | 13.79 | 0.331 | 0.468 |
| Mip-Splatting | 13.70 | 0.315 | 0.486 |
| **Ours** | **15.51** | **0.379** | **0.465** |

perceptual and textural information in NVS outputs. Compared to the original VAE model, our fine-tuning approach significantly enhances 3D-consistency in the 2D latent representations by enforcing the correspondence points to be consistent, resulting in superior latent NVS performance across all metrics. "Image Space" means that we input images to 3DGS with the same resolution as the latent representations, then render output images with the same resolution as before latent encoding. Since we render high-resolution images from low-resolution training images, to avoid unfair comparisons caused by aliasing, we also compare our method with the Mip-Splatting (Yu et al., 2024) which is specialized at super-resolution rendering. Compared with these image space methods, our latent reconstruction method still achieves better performance on most of the datasets, highlighting its potential for future work in efficient 3D representation learning.

We also show the generalizability by performing the synthesis of few-shot novel views in NeRF-LLFF dataset. We follow the same experimental configurations as in the previous work (Li et al., 2024; Liu et al., 2024). And we keep the same input resolution for all the methods. As shown in Table 2, our method outperforms the other image-space approaches in the sparse-view setting.

## 5.2 TEXT-TO-3D GENERATION

We evaluate our method for the state of art text-to-3D generation framework in both latent and image space. We leverage the GSGEN (Chen et al., 2024) and Dreamfusion (Poole et al., 2022) as the image space generation framework, while we use Latent-NeRF (Metzer et al., 2022) as the latent space method. GSGEN is optimized in the 512×512 image space. Dreamfusion is optimized in the 800×800 image space. Latent-NeRF is optimized in the 128×128 latent space and then reconstruct images to a resolution of 1024×1024. By following the prompts evaluated in these two works, we generate 3D objects and render them from multiple views. The text prompts fed into the GSGEN are more complicated considering it is the state of the art generation method.

As shown in Fig. 5, our method can boost the performance under extremely complicated text prompts, achieve complex geometry while preserving the multi-view consistency. Moreover, our encoder model can significantly enhance the high-frequency details such as the texture of the fried chicken. Besides, our approach is compatible with the diffusion model operating within the original VAE latent space. Without necessitating any fine-tuning of the diffusion U-Net parameters, the diffusion process remains capable of accurately denoising the 2D latent representations provided by our fine-tuned VAE, according to the text guidance. Furthermore, the VAE-RF alignment in decoder

Table 3: We ablate correspondence-aware autoencoding and VAE-radiance field aligned decoder fine-tuning on DL3DV-10K dataset to reveal their necessity in latent 3D reconstruction .

| VAE | Encoder fine-tuned | Decoder fine-tuned | PSNR↑ | SSIM↑ | LPIPS↓ |
|:---:|:---:|:---:|:---:|:---:|:---:|
| ✓ | - | - | 20.57 | 0.595 | 0.346 |
| ✓ | ✓ | - | 21.16 | 0.620 | 0.282 |
| ✓ | - | ✓ | 21.73 | 0.645 | 0.208 |
| ✓ | ✓ | ✓ | **22.45** | **0.667** | **0.197** |

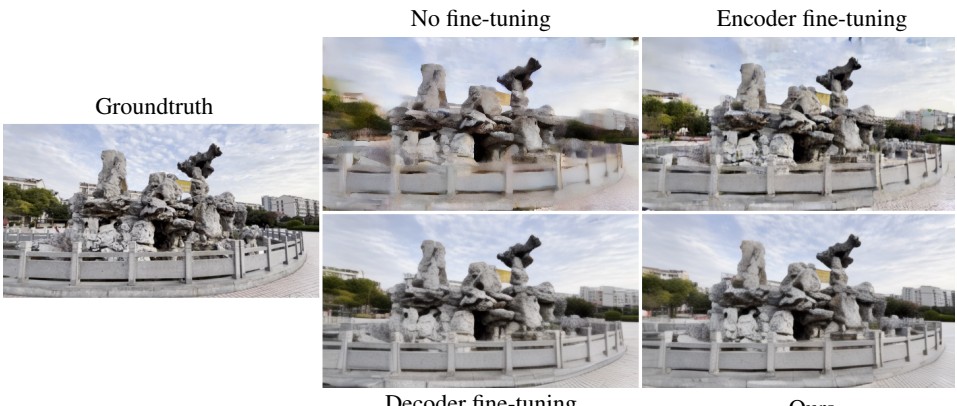

Figure 6: A qualitative study of the effect of different fine-tuning stages for view synthesis results.

fine-tuning also facilitates the reconstruction of rendered latent representations, improving the image quality after VAE decoding.

## 5.3 ABLATION STUDY

We conduct ablation studies on two major components of our three-stage framework, the correspondence-aware autoencoding and the VAE-RF aligned decoder fine-tuning, to assess their contributions to overall performance. The quantitative results shown in Table 3 indicate that both components contribute to performance improvement. Notably, the decoder presents a more significant impact on the results, as it directly influences the reconstruction of images from the latent space, thereby leading to stronger performance gains. Although the encoder does not directly act on image reconstruction, it enhances geometric consistency of 2D representations, which also contributes to the performance improvement in 3D reconstruction.

The qualitative results are shown in Fig. 6. The encoder fine-tuning allows the 3D latent space to capture more precise geometry, reduce blurriness in the synthesized images, and recover finer details. Additionally, the decoder fine-tuning further refines the results by rectifying inaccuracies and preserving perceptual and textural fidelity. Together, these modules synergistically contribute to significant improvements in the overall pipeline.

## 6 CONCLUSION

This paper introduces the Latent Radiance Field (LRF), which to our knowledge, is the first work to construct radiance field representations directly in the 2D latent space for 3D reconstruction. We present a novel framework for incorporating 3D awareness into 2D representation learning, featuring a correspondence-aware autoencoding method and a VAE-Radiance Field (VAE-RF) alignment strategy to bridge the domain gap between the 2D latent space and the natural 3D space, thereby significantly enhancing the visual quality of our LRF. Future work will focus on incorporating our method with more compact 3D representations, efficient NVS, few-shot NVS in latent space, as well as exploring its application with potential 3D latent diffusion models.

ACKNOWLEDGMENT

The authors thank Minghui Xu for beneficial discussions. This work is partially supported by the AIM for Composites, an Energy Frontier Research Center funded by the U.S. Department of Energy (DOE), Office of Science, Basic Energy Sciences (BES), under Award # DE-SC0023389 and by the US National Science Foundation (NSF; Grant Number MTM2-2025541, OIA-2242812). The authors acknowledge research support from Clemson University with a generous allotment of computation time on the Palmetto cluster.

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

# A   APPENDIX

## A.1   DETAILS OF CALCULATING THE WEIGHT

To compute $\lambda_{ij}$, we first calculate the Absolute Pose Error (APE) for each pose pair using the formula: $E_{ij} = P_i^{-1} P_j$, where $P_i$ and $P_j$ are the different camera poses respectively. After obtaining $E_{ij}$, the APE is calculated as:

$$APE_{ij} = \|E_{ij} - I_{4\times4}\|_F, \tag{12}$$

where $I_{4\times4}$ is the identity matrix $F$ and represents the Frobenius norm. In each iteration, the APE values are normalized across all image pairs to derive the weights $\lambda_{ij}$, as: $\lambda_{ij} = \frac{APE_{ij}}{\sum_k APE_k}$, where $k$ represents each image pair within one iterations. This normalization ensures they reflect the relative contributions of each pose error in a consistent manner. This method is implemented based on the APE computation approach in the evo library (Grupp, 2017).

## A.2   DETAILS OF DATASET

We create a correspondence pair dataset based on the training set of DL3DV-10K (Ling et al., 2024) dataset to fine-tune our VAE encoder. We randomly sample 784 scenes and extract correspondence pairs from the multi-view images by using COLMAP. The correspondence points for each scene will be pre-computed before the model fine-tuning process. We use a sequential matcher with the number of overlapping images set to 10 and the number of quadratic overlaps set to 1. Such overlapping searching strategy ensures our model not only learns from easy and dense correspondence, but also from challenging cases among far-view image pairs, adding great robutness for our model. The ability to remain consistency in large view difference is particularly necessary for the outdoor unbounded reconstruction. Moreover, we set the minimum number of inliers and minimum ratio of inliers to 15 and 0.25 with the loop detection to make sure the extracted correspondence is accurate enough. We also train the same number of latent 3D Gaussian splatting scenes from the DL3DV-10K datasets to create a paired dataset of images and rendered latents, which are used for Stage-III decoder fine-tuning.

## A.3   IMPLEMENTATION DETAILS

For Stage-I, we employ the pre-trained VAE model ($f = 8$, $KL$), from LDM model zoo as the backbone VAE model. We fine-tune the VAE on 2 NVIDIA A100-80GB GPUs for around one day, by using the correspondence pair dataset with an image resolution of $512\times512$, the base learning rate of $4.5e - 06$, and the default optimizer. For Stage-III, we fine-tune the decoder on the image-latent dataset with 2 NVIDIA A100-80GB GPUs for around one day.

In the implementation of LRF, we normalize the latent input to the radiance field using the scale of all input views to stabilize radiance field optimization, and apply denormalization during rendering. During the VAE encoding stage, we start the discriminator at step 501 for better image quality, and we set $KL_{\text{weight}} = 1.0 \times 10^{-6}$, and $\mathcal{D}_{\text{weight}} = 0.5$. For the decoder training, we use the same configuration as the original VAE, except $KL_{\text{weight}} = 0$ to ensure only the decoder was optimized.

## A.4   IMAGE RECONSTRUCTION PERFORMANCE

To verify that our approach does not degrade the performance on downstream tasks, we evaluate the image reconstruction performance of our fine-tuned VAE by calculating PSNR between the original images and the reconstructed images. As shown in Table 4, adding the correspondence consistency constraint to inject 3D awareness and applying a regularization loss to keep the latent space close to the original latent space perform minimal impact on the VAE's reconstruction performance. This ensures that our VAE model can still be effectively used in conjunction with other pre-trained models, such as the Stable Diffusion model, without any fine-tuning.

Table 4: Evaluation of PSNR for images reconstructed by VAEs on NeRF-LLFF, DL3DV-10K, and Mip-NeRF360 datasets.

| Method | Metric | NeRF-LLFF | DL3DV-10K | Mip-NeRF360 |
|---|---|---|---|---|
| **VAE** | PSNR ↑ | 23.47 | 24.59 | 24.54 |
| **Our-VAE** | PSNR ↑ | 23.59 | 23.25 | 24.24 |

## A.5 MORE IMAGE GENERATION RESULTS

Fig. 7 demonstrates that our VAE model can generate 3D objects guided by text prompts without any fine-tuning of the diffusion model. Moreover, Fig. 8 shows that our VAE can also improve the GSGEN (Chen et al., 2024) to achieve better 3D generations with complicated text prompts.

## A.6 EFFICIENCY ANALYSIS

Table 5 demonstrates that our method reduces input resolutions, model storage space, and GPU usage for photorealistic NVS, which is particularly useful in cases with limited communication bandwidth and storage. For instance, some individuals may not have GPUs with large memories, where our method is an efficient solution for them to run photorealistic NVS algorithms.

Table 5: Efficiency comparison of different image-space and latent-space NVS methods.

| Method | Input resolution | Training Time ↓ | GPU Usage ↓ | Storage ↓ | Rendering FPS ↑ | Decoding FPS ↑ | PSNR ↑ | SSIM ↑ | LPIPS ↓ |
|---|---|---|---|---|---|---|---|---|---|
| *3DGS* | *512×512* | *5.9 min* | *3 GB* | *200.41 MB* | *100* | *-* | *26.17* | *0.778* | *0.009* |
| 3DGS/8 | 64×64 | **3.1 min** | 1 GB | **59.15 MB** | **200** | - | 14.03 | 0.352 | 0.541 |
| 3DGS-VAE | 64×64 | 4.8 min | 2 GB | 250.97 MB | 80 | 20 | 20.57 | 0.595 | 0.346 |
| Latent-NeRF | 64×64 | 27.2 min | 10 GB | 350.50 MB | 0.09 | 20 | 18.16 | 0.530 | 0.432 |
| Ours | 64×64 | 3.9 min | **1 GB** | 96.42 MB | 180 | 20 | **22.45** | **0.667** | **0.197** |

## A.7 MORE EXPERIMENTAL RESULTS

To demonstrate the effectiveness and generalizability of our method for 3D latent reconstruction, we show more NVS and 3D generation results on four datasets covering indoor scenes, outdoor scenes, and object-level scenes. As shown in Fig. 9, 10, 11 and 12, our method yields a significant improvement in image quality.

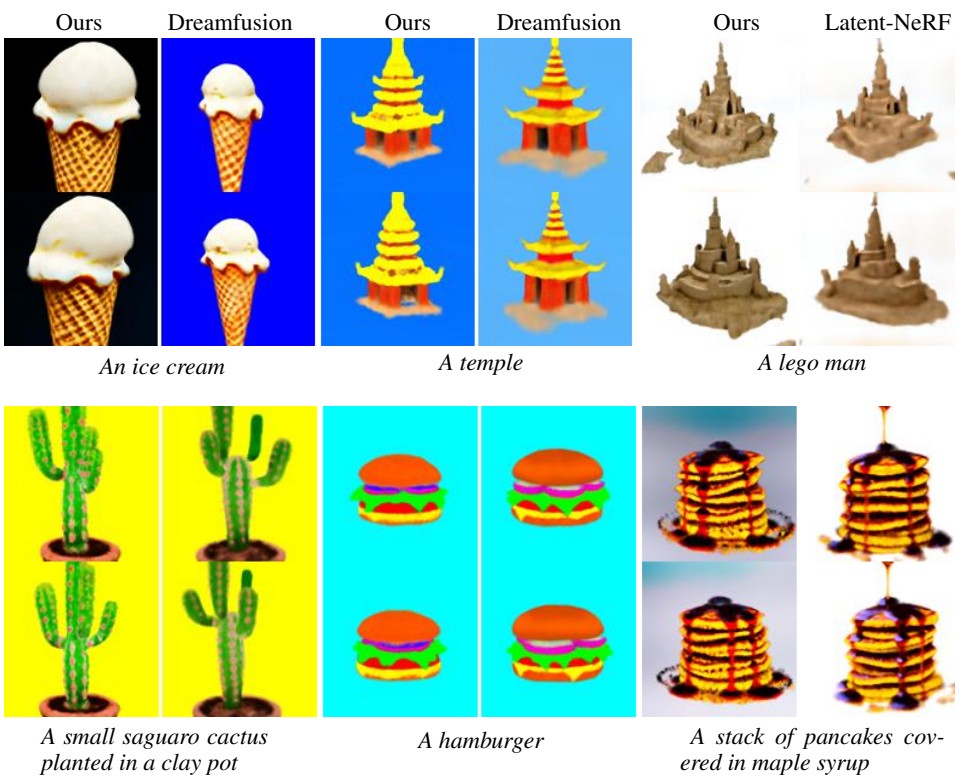

Figure 7: Samples for text-to-3D generation on the image and latent space.

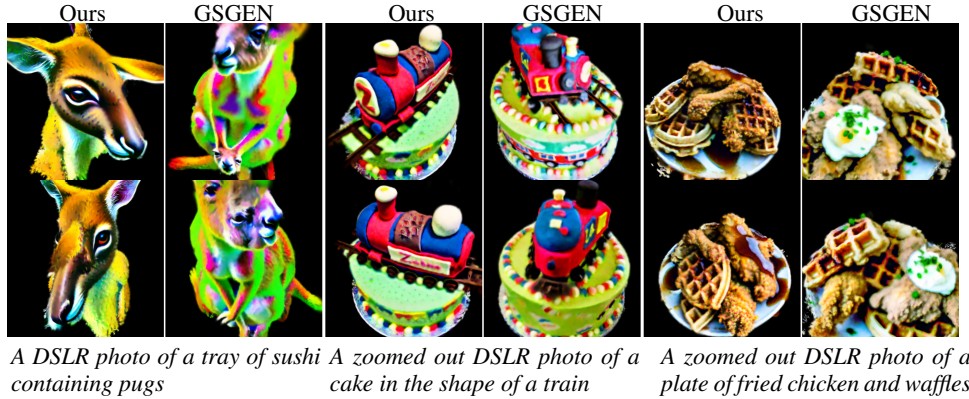

Figure 8: More samples for text-to-3D generation on the image space.

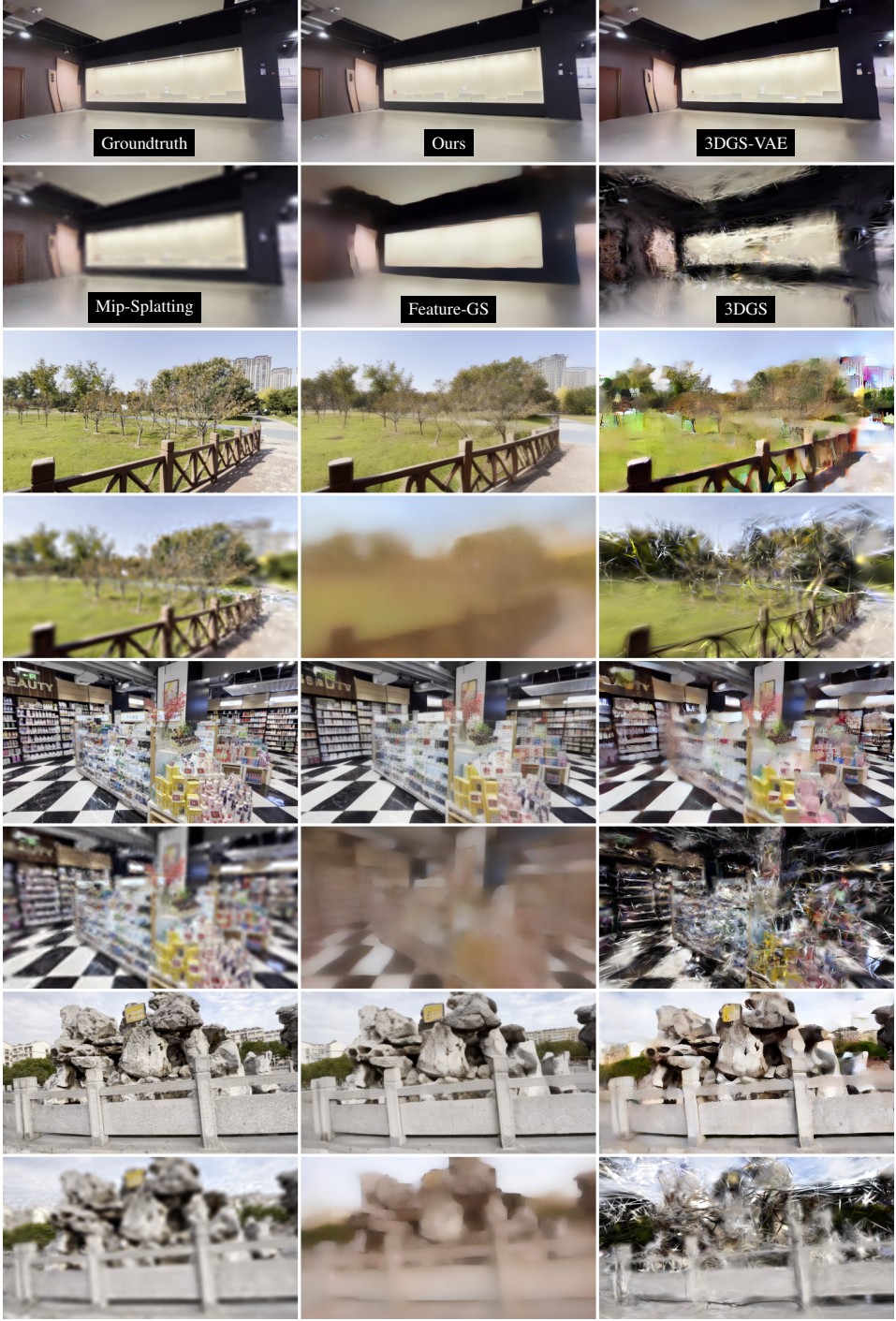

Figure 9: More NVS results on the **DL3DV-10K** dataset.

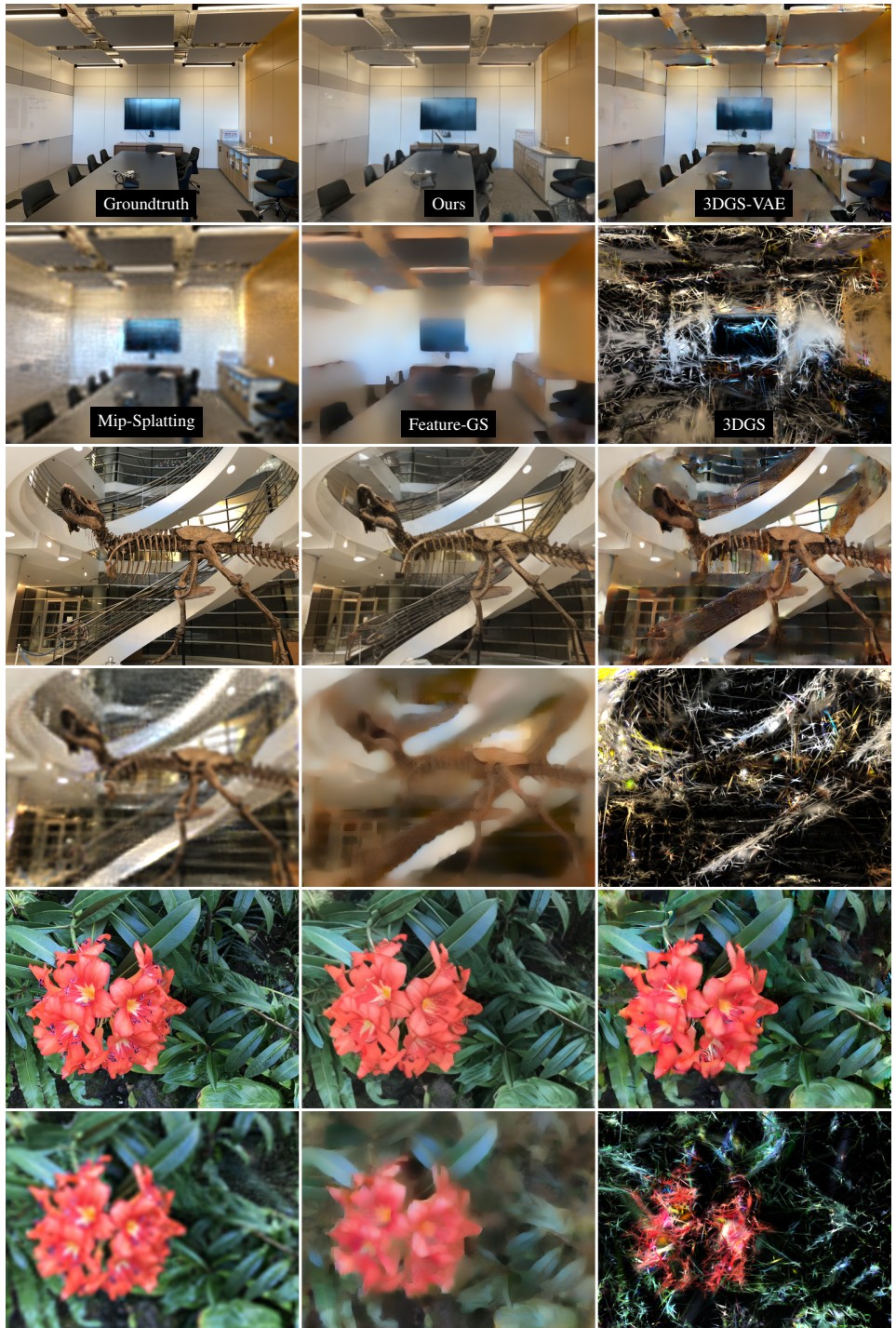

Figure 10: More NVS results on the **NeRF-LLFF** dataset.

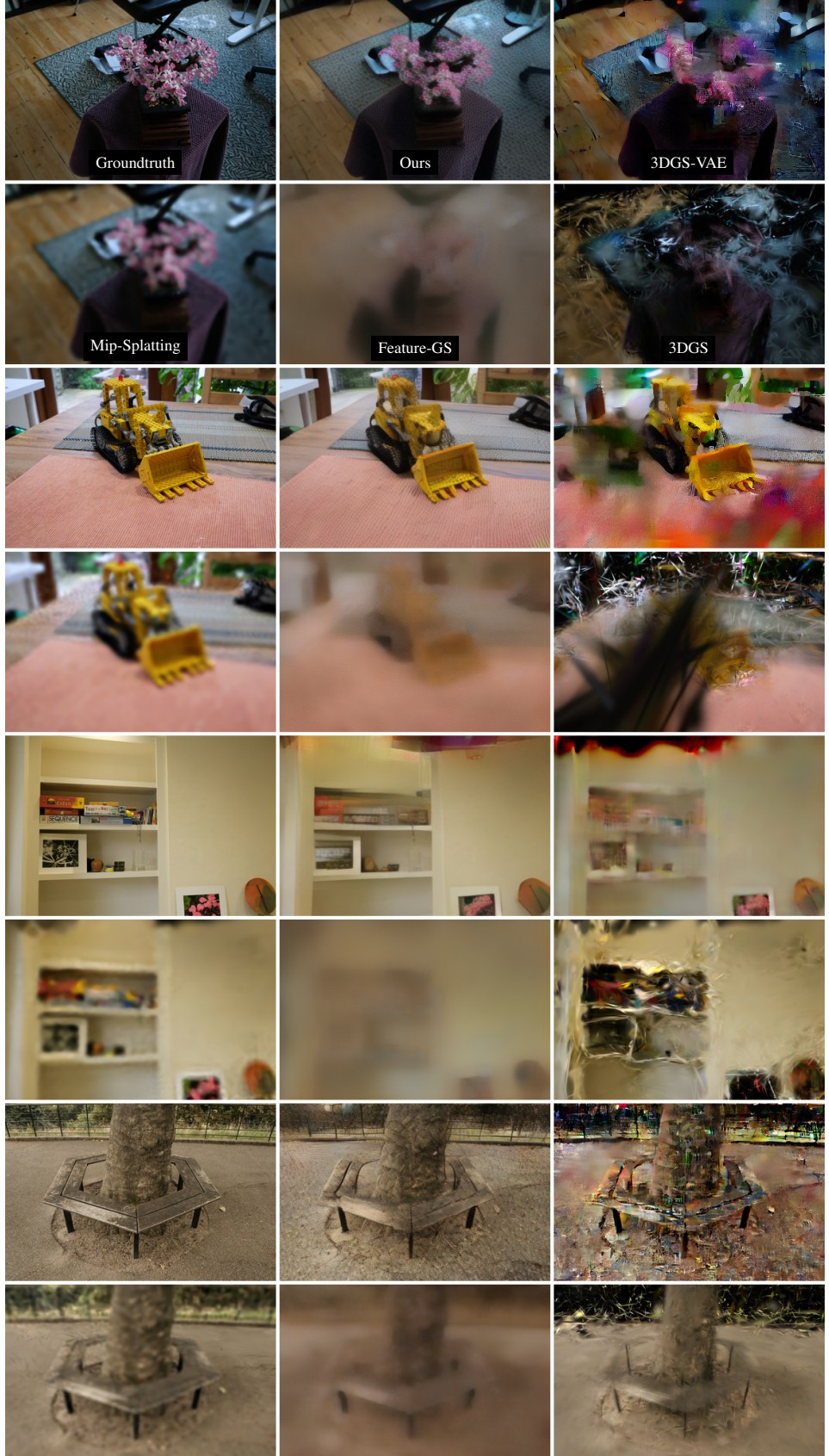

Figure 11: More NVS results on the **Mip-NeRF360** dataset.

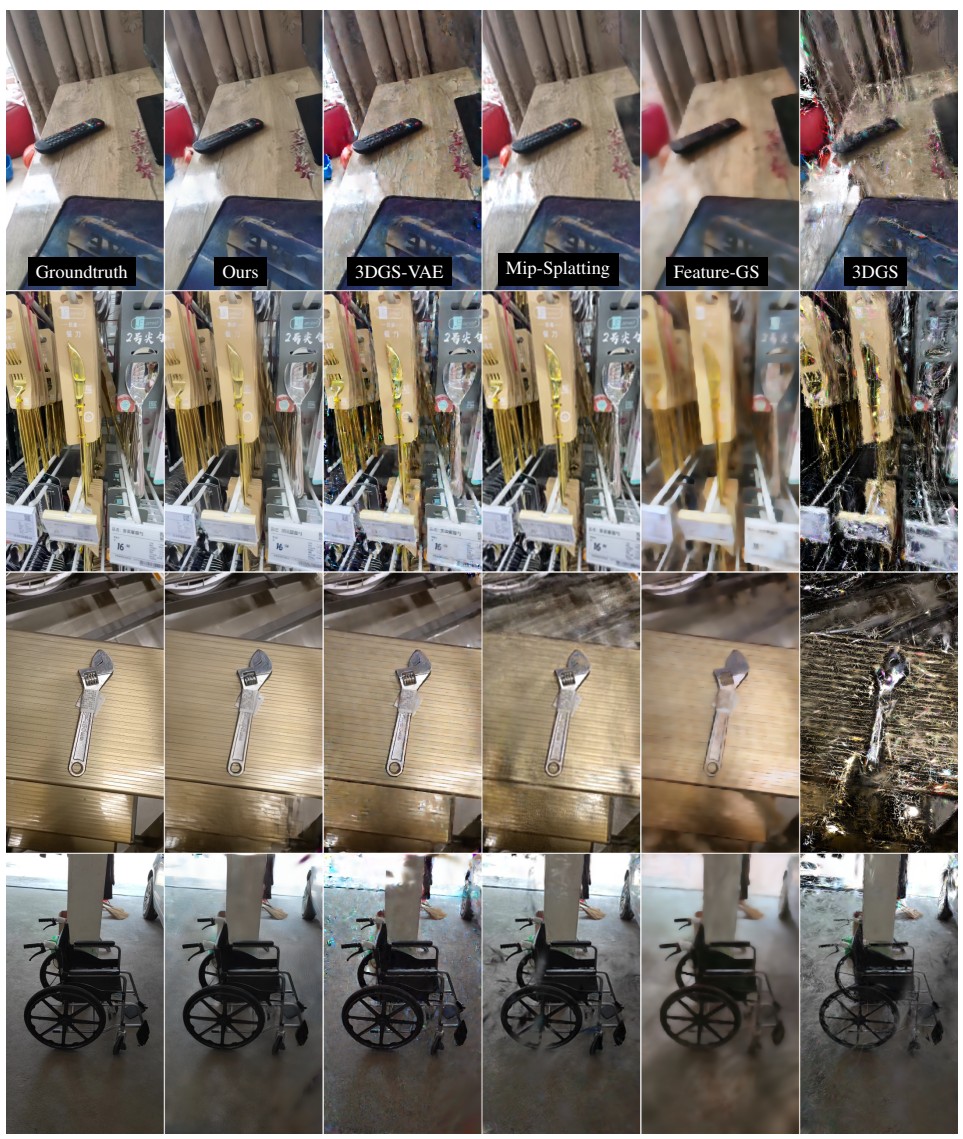

Figure 12: More NVS results on the **MVImgNet** dataset.

