# OpenReview forum: "Latent Radiance Fields with 3D-aware 2D Representations"
_ICLR.cc/2025/Conference — ICLR 2025 Poster_

### Official Review · Reviewer_Cwtt · 2024-10-26

**Soundness:** 3
**Presentation:** 3
**Contribution:** 2
**Rating:** 6
**Confidence:** 5

**Summary:**

This paper target at Latent 3D reconstruction, and address the domain gap between 2D feature space and 3D representations. They proposed propose a novel framework that comprise (1) a correspondence-aware autoencoding method, (2) a latent radiance field (LRF), and (3) a VAE-Radiance Field (VAE-RF) alignment strategy.

**Strengths:**

Strengths*:
With the proposeld framework, this paper enhances the 3D consistency of 2D latent representations as well as effectively mitigated the gap between the 2D latent space and the natural 3D space.

**Weaknesses:**

Weaknesses*:

1. Compared with feature-GS, it is a good improvement to add a correspondence-aware constraint during VAE encoder finetuning to improve its 3D awareness. However, this approach still cannot guarantee strict multi-view consistency of the encoded multiview features. As a result, after constructing the LRF, there may be a blurred radiance field with significant detail losses. Although the LRF is 3D consistent, the final decoded features may still exhibit noticeable flickering effects due to the lack of view consistency of decoder.


2. I think this paper still needs optimization during the LRF stage instead of using a total feed forward method. When compared with 3DGS and Mip-Splatting, the authors only train them in latent space resolution (8 times lower than the image resolution), which yields very bad visual results. I suggest that the authors consider training the competing methods at full resolution. This paper may not work better than Mip-Splatting when trained with full resolution and very dense view, but it would be interesting to see if it outperforms Mip-Splatting in a sparse-view setting. This might be achieved because of generation capability of Stable Diffusion VAE used in this paper.

3. The author didn't provide a detailed explanation of experiment settings. For example, I wonder how views are sampled in the experiment during training. And how many views are used as input and evaluation repectively. This is important for me to fully evaluate this paper.

4. Object change in final rendering. As shown in Fig 1, the building in the final rendering is different from the ground truth image. One is in red, the other one is in white. This lead to my concern of identity preserving capability of purposed method. I think this is a problem that needs to be addressed.

**Questions:**

I think the strict view-inconsistency can not be solved fundamentally due to the emplyment of VAE Decoder. But it coule be possible to showcase more view consistent results and evlaute the 3D consistency of proposeld methods.

The author provide more details about the experiment settings, especially the view sampling strategy and the number of views used in the experiment. I hope see whether the proposed method can be applied to a sparse-view setting.

---

> ### Author Response · Authors · 2024-11-21
>
> Thank you very much for your time and effort in reviewing our work and providing the constructive comments.
>
> **W1. Decoder finetuning**:
> Thank you for the valuable comment. The view inconsistency cannot be fully addressed by the encoder fine-tuning. However, with our fine-tuned encoder, the latent radiance field can be built more effectively, which serves as a foundation for further fine-tuning the decoder for better view inconsistency in the decoding process.
>
> Current view-consist decoding methods often leverages temporally consistent decoders (e.g., 3D VAEs) [r1, r2, r3] to achieve view-consistent decoding. However, pre-trained 3D VAEs typically do not outperform standard VAEs in terms of performance; their primary advantage lies in computational efficiency. While fine-tuning these 3D VAEs can improve view consistency, they remain ineffective for handling large view gaps (often in our setting) due to the limitations of their Nx1x1 temporal kernel size.
>
> In contrast, our approach focuses on encoder fine-tuning, which inherently enables our decoder to achieve view consistency without requiring substantial adjustments. Consequently, this work took a more general approach for decoder fine-tuning to demonstrate the effectiveness of our encoder fine-tuning.
>
> [r1] CV-VAE: A Compatible Video VAE for Latent Generative Video Models
>
> [r2] Upscale-A-Video: Temporal-Consistent Diffusion Model for Real-World Video Super-Resolution
>
> [r3] CogVideoX: Text-to-Video Diffusion Models with An Expert Transformer
>
> **W2. Trying sparse views.**:
>
> Thank you for your very insightful suggestion! The table below demonstrates that our method indeed outperforms the other image-space approaches in the sparse-view setting (3 views) on the LLFF dataset. Our method is even better than the state-of-the-art 3DGS methods with standard input image resolutions.
> | Method (LLFF dataset, 3 views)                                     | PSNR &uarr; | SSIM &uarr; | LPIPS &darr; |
> |-------------------------------------------|-------------|-------------|--------------|
> | 3DGS-VAE                                  | 13.06       | 0.283       | 0.5708       |
> | 3DGS (standard resolution)                | 13.79       | 0.331       | 0.468        |
> | Mip-Splatting (standard resolution)       | 13.70       | 0.315       | 0.486        |
> | Ours                                      | **15.51**   | **0.379**   | **0.465**    |
>
>
> **W3. Train/test split**:
> We split the training and test data by selecting camera poses whose indices are divisible by 8 as the test dataset, following the same train/test split setting of 3D Gaussian splatting.
>
> **W4. Color shift in teaser**:
>
> Thank you for pointing this out. There are still noticeable color shifts (i.e., the distribution shifts discussed in Section 3.3) in the novel view latent rendering. In general, our method has addressed most of the color shift issues. As shown in Figure 6, comparing 3DGS-VAE and our method, there is a significant improvement in color consistency. However, some inevitable color shifts remain due to the imperfection of applying RGB-based NVS pipeline to latent features. The SHS faces challenges in effectively handling view-dependent latent representations.
>
> **Q1. More view-consistent visual results**:
> We evaluate the 3D consistency of different latent representations by constructing 3DGS models in the latent space. A higher PSNR in NVS tasks indicates better 3D consistency, and our method achieves a higher PSNR compared to the vanilla VAE, as shown in Figure 6 and Table 2 of the paper. Please visit our project website https://latent-radiance-field.github.io/LRF for videos showing better consistency between video frames (i.e., different views) rendered by our method.
>
> **Q2. Train/test split**:
> Please refer to our response to W3.

---

> > ### Comment · Reviewer_Cwtt · 2024-11-24
> >
> > Although Color shift and View consistency can not be addressed perfectly, I still could raise my score to 6 due to the added sparse view experiment. Because it outperforms Mip-Splatting in a sparse-view setting using the same resolution, showing the generation capability of Stable Diffusion VAE as well as the capability of 3D preserving of the proposed method.

---

> ### Author Response · Authors · 2024-11-24
>
> Thank you again for your time and effort in reviewing our work and providing the constructive comments.

---

### Official Review · Reviewer_RdaN · 2024-11-02

**Soundness:** 2
**Presentation:** 3
**Contribution:** 3
**Rating:** 5
**Confidence:** 4

**Summary:**

This paper introduces a framework for constructing radiance field representations in latent space, aiming to bridge the domain gap between 2D feature space and 3D representations. The authors propose a three-stage pipeline: (1) a correspondence-aware autoencoding method that enforces 3D consistency in latent space through correspondence constraints, (2) a latent radiance field (LRF) that lifts these 3D-aware 2D representations into 3D space, and (3) a VAE-Radiance Field alignment strategy that improves image decoding from rendered 2D representations.

The key technical contribution is the integration of 3D awareness into 2D representation learning without requiring additional per-scene refinement modules. The authors adapt the 3D Gaussian Splatting framework to operate in latent space, using spherical harmonics to model view-dependent effects. They demonstrate their method's effectiveness on both novel view synthesis and text-to-3D generation tasks across various datasets including MVImgNet, NeRF-LLFF, MipNeRF360, and DL3DV-10K. The authors claim their approach is the first to achieve photorealistic 3D reconstruction performance directly from latent representations while maintaining cross-dataset generalizability.

The work represents an attempt to make latent 3D reconstruction more practical by addressing the geometric consistency issues in existing approaches. The framework is designed to be compatible with existing novel view synthesis and 3D generation pipelines without requiring additional fine-tuning.

**Strengths:**

- The paper follows a standard pipeline structure addressing latent space 3D reconstruction. The method section breaks down into three components: correspondence-aware encoding, latent radiance field construction, and VAE alignment. The ablation study provides basic validation of these components, though more comprehensive analysis would be beneficial.
- While building heavily on existing techniques, the paper demonstrates competent engineering in combining different elements into a working system. The adaptation of correspondence constraints and 3DGS to latent space shows reasonable technical implementation. The provided implementation details outline the basic approach.
- The evaluation includes tests on multiple datasets (MVImgNet, NeRF-LLFF, MipNeRF360, DL3DV-10K), attempting to demonstrate applicability across different scenarios. While the cross-dataset evaluation has limitations, it provides basic evidence of generalization capability. The inclusion of both novel view synthesis and text-to-3D generation shows the method's potential utility, though more thorough evaluations are needed.
- The method functions without per-scene refinement modules, which could be advantageous compared to some previous approaches.

**Weaknesses:**

- The paper fails to provide compelling justification for operating in latent space. While previous works like Latent-NeRF (for text-to-3D generation) established initial groundwork, this paper does not clearly demonstrate additional benefits of its approach. The motivation for operating in latent space remains questionable. The paper shows modest improvements in PSNR/SSIM metrics but doesn't address fundamental questions: What are the computational advantages over image-space methods? How does memory consumption compare? Why is the added complexity of latent space operations justified? The authors should conduct a thorough efficiency analysis, measuring training time, inference speed, and memory usage against image-space baselines. Without such evidence, the practical value of the latent space approach is hard to justify.
- Section 4.1's correspondence mechanism has several fundamental issues. Most critically, the paper fails to address the scale mismatch between COLMAP's pixel-level correspondences and the VAE's latent space. Given that the VAE operates at a lower resolution (likely 8x or 16x downsampled) with larger receptive fields, how are pixel-level correspondences meaningfully mapped to latent features? This mapping is non-trivial: a single latent code typically corresponds to a large receptive field in pixel space, making precise correspondence matching questionable. The paper should answer: How are multiple pixel correspondences within one latent cell handled? How does the receptive field size affect correspondence accuracy? Additionally, basic details are missing: the correspondence filtering criteria, quality metrics, and robustness to matching errors. The use of L1 distance for latent features (Eq. 6) needs justification, especially given the coarse nature of latent correspondences. These technical gaps raise serious concerns about the method's fundamental soundness.
- The use of spherical harmonics in latent space (Eq. 8) is puzzling. Given that the features are already in a learned latent space, why introduce SH basis functions? A direct learnable decoder or simpler view-dependent representation might suffice. Similarly, the VAE-RF alignment stage seems unnecessarily complex - the authors may quantify the alleged distribution shift and explore simpler alternatives. These design choices add complexity without clear benefits.
- The experimental setup has a fundamental flaw: image-space methods are handicapped by low-resolution inputs while the proposed method has access to high-resolution data. This creates an artificial advantage for the proposed method. A fair comparison requires either: testing at matched resolutions, or demonstrating specific benefits under computational constraints. The ablation study skips crucial experiments on correspondence quality, loss function components, and architectural variations. These missing comparisons make it difficult to assess the true value of each component.
- The paper sidesteps important practical concerns. Where are the failure cases? How does the method handle challenging scenes with varying illumination or complex geometry? The text-to-3D generation results lack comparisons with current state-of-the-art methods. The claim of "photorealistic reconstruction" needs validation through proper user studies or established perceptual metrics. Testing on more diverse, challenging scenarios would better demonstrate real-world applicability.

**Questions:**

- How to handle the scale mismatch between COLMAP and latent space? Please clarify: 1) the exact mapping strategy from pixel to latent correspondences; 2) how multiple pixel correspondences within one latent cell are aggregated
- What's the correspondence filtering pipeline? In particular: 1) thresholds used for COLMAP matching 2) any additional filtering criteria in latent space 3) how outlier correspondences are handled.
- During VAE-RF alignment (Section 4.3), how to: 1) balance the training/novel view losses 2) prevent overfitting during decoder fine-tuning.
- Regarding the resolution setup, why choose this specific resolution comparison protocol? Would the method's advantages hold at equal resolutions?

---

> ### Author Response · Authors · 2024-11-21
>
> Thank you very much for your time and effort in reviewing our work and providing the constructive comments.
>
> **W1. What are the computational advantages over image-space methods?**:
>
> Please refer to the following figure with the details about the training time, GPU usage, storage, and rendering speed. Our method reduces input resolutions, model storage space, and GPU usage for photorealistic NVS, which is particularly useful in cases with limited communication bandwidth and storage.
>
> | Method           | Input resolution to 3D model | Training Time &darr; | GPU Usage &darr; | Storage &darr;       | Rendering FPS &uarr; | Decoding FPS &uarr; | PSNR &uarr; | SSIM &uarr; | LPIPS &darr; |
> |------------------|-------------------------|-----------------------|------------------|----------------------|----------------------|---------------------|-------------|-------------|--------------|
> | *3DGS*           | *512\*512*              | *5.9 min*             | *3 GB*           | *200.41 MB*          | *100*                | *-*                 | *26.17*     | *0.778*     | *0.009*      |
> | 3DGS/8           | 64*64                  | **3.1 min**               | 1 GB             | **59.15 MB**         | **200**              | -                   | 14.03        | 0.352       | 0.541        |
> | 3DGS-VAE         | 64*64                  | 4.8 min               | 2 GB             | 250.97 MB            | 80                   | 20                  | 20.57       | 0.595       | 0.346        |
> | Latent-NeRF      | 64*64                  | 27.2 min              | 10 GB            | 350.50 MB            | 0.09                 | 20                  | 18.16       | 0.530       | 0.432        |
> | Ours             | 64*64                  | 3.9 min           | **1 GB**         | 96.42 MB             | 180                  | 20                  | **22.45**       | **0.667**       | **0.197**        |
>
> **W2.1. Scale mismatch**:
>
> Thank you for the insightful comment. We have studied the proposed correspondence loss at multiple spatial levels by applying it to different downsampling layers of the VAE encoder. The performance gains were found to be trivial. To balance computational efficiency and performance, we decided to apply correspondence loss only to the output feature maps of VAE encoder.
>
> **W2.2. COLMAP details**:
>
> Regarding the setting of COLMAP, the correspondence points for each scene will be pre-computed before the model fine-tuning process. We use sequential matcher with the number of overlapping images of 10 and number of quadratic overlap of 1. Such overlapping searching strategy ensures our model not only learns from easy and dense correspondence, but also from challenging cases. Moreover, we set the minimum number of inliers and minimum ratio of inliers to 15 and 0.25 with the loop detection to make sure the extracted correspondence is accurate enough. Even though our correspondence calculation is robust, more ablation studies will be added in the camera ready revision to investigate the performance of different noise levels of correspondence points to reveal the impact of outlier correspondence.
>
> **W3.1. Why introduce SH basis functions? A direct learnable decoder or simpler view-dependent representation might suffice**:
>
>  Thank you for the great insight. We agree that per-scene view-dependent layer may also work, but it requires much more model optimization time and rendering time compared to SHS.  In this work, we need to build a large dataset of latent radiance fields for 1K scenes. Therefore, we chose SHS as the 3D representations for an efficient optimization.
>
> **W3.2. Distribution shift in latent representations**:
>
>  We observed that there are distribution shifts between latent features rendered from 3D representations and encoded by VAEs. To verify this, we calculate KL divergence to quantify the distribution shift before and after novel view synthesis by LRF.
> | KL Loss (VAE encoder output)  | KL Loss (after 3DGS rendering)  |
> |------------------------------|----------------------------|
> | $3.35*10^4$                    |          $1.32*10^5$       |
>
>
> Please also refer to Fig. 6 of the paper, where no-decoder-finetuning results in worse visual quality. These two evidences indicate the necessity of fine-tuning the decoder.
>
> **W4. Comparison protocol**:
> The purpose of comparing our method to 3DGS with 8x downsampled input resolution is to evaluate the performance of 3DGS models under the same input resolutions. When the latents share the same resolution as the RGB images, the 3D latent representation outperforms all existing methods of 3D RGB representation. This finding provides valuable insights for downstream tasks requiring latent representations while addressing limited computational constraint.

---

> ### Author Response · Authors · 2024-11-21
>
> **W5. More recent text-to-3D generation**:
> We have conducted comparison with more recent text-to-3D generation method, as shown in Sec. 5.3 of the revised paper. As shown in Fig. 5, our method can boost the performance under extremely complicated text prompts, and achieve complex geometry while preserving the multi-view consistency. For instance, in the result of GSGEN, the train on the cake appears distorted and inconsistent shapes across views. Our method achieves consistent generation across views.
>
> **Q1. Correspondence point mapping**:
>
> The mapping is performed by grid sampling, where pixel-level correspondences are mapped to the latent space through bilinear interpolation. This ensures that each latent position receive correspondence information from its associated pixels accurately.
>
>
> **Q2. COLMAP settings**:
>
> Please refer to our response to W2.2.
>
> **Q3.1. How to balance the weight**:
> We set both $\lambda_{\text{train}}$  and $\lambda_{\text{novel}}$ to 0.5. Our dataset consists of  80% training view images and 20% novel view images. By assigning equal weights to these terms, we ensure that the decoder learns not only to decode effectively from the training views but also to generalize and perform well on the novel views.
>
> **Q3.2. Over-fitting risk**:
> Thank you for the valuable comment. To avoid over-fitting of VAE fine-tuning, we held a validation set from the 1K scenes and selected the checkpoint with the best performance on the validation set. Additionally, we have performed extensive out-of-domain evaluations on unseen datasets such as NeRF-LLFF, MVImageNet, and MipNeRF 360 to demonstrate the strong generalization ability of our approach.
>
> **Q4. Comparsion Protocol**:
> Please refer to our response to W4.

---

> > ### Comment · Reviewer_RdaN · 2024-11-24
> > **Thanks for the rebuttal**
> >
> > Thank you for the detailed response. However, there remain concerns about the fairness of the experimental comparison protocol that have not been adequately addressed in the rebuttal.
> >
> > As demonstrated in the first Table, when the 3DGS baseline has access to original resolution inputs (512×512), it significantly outperforms the proposed method in novel view synthesis metrics. This highlights a fundamental issue with the current comparison methodology: the proposed approach has access to high-resolution images during the encoding phase, while baseline image-space methods are restricted to low-resolution inputs only. This creates an inherent advantage that makes the comparisons methodologically problematic.
> >
> > While the presented method demonstrates benefits in terms of reduced memory consumption and faster training time, it is worth noting that the original 3DGS baseline is already quite practical and can run on most consumer-grade GPUs. Moreover, the proposed approach requires additional pre-training overhead, which should be factored into the total computational cost analysis.
> >
> > It would be valuable to either conduct experiments where all methods have access to the same resolution inputs, or explicitly reframe the work's contributions to focus on computational efficiency rather than reconstruction quality. The paper should clearly acknowledge these limitations and trade-offs, particularly the resolution access disparity in the experimental setup. In my opinion, the current comparison protocol impacts the validity of the paper's claims. A more transparent presentation of these limitations would strengthen the paper's contribution to the field and help readers better understand the true advantages and trade-offs of the proposed approach.

---

> ### Author Response · Authors · 2024-11-27
>
> Thank you for the further comments. We would like to reemphasize the motivation of this work, as to *"bridge the gap between 2D and 3D representations, providing an efficient and generalizable solution for tasks requiring multi-view consistency in the feature space"*. Targeting at this, our focus on 3D reconstruction is **feature-space reconstruction** rather than image-space reconstruction. Compared with the feature-space reconstruction baselines such as Feature 3DGS and Latent-NeRF, our method achieves better reconstruction performance, demonstrating the effectiveness of introducing 3D awareness into the 2D feature space.
>
> **The gap between feature-space reconstruction and full-resolution image-space 3DGS** is mainly due to the image quality degradation incurred by image VAE reconstruction. As evidenced in the table below, reconstructing images by VAE in the 2D space (without 3D reconstruction) only maintain 24.59 PSNR, degrading from the infinite PSNR (i.e., comparing two identical images). This serves as the upper-bound of feature-space reconstruction methods. Our method conducts 3D reconstruction in the feature space while approaches this upper-bound with a 22.45 PSNR. The image-space methods shown in the paper, like 3DGS and Mip-Splatting, serve as a reference to feature-space methods while reveal an insight:
> - *Under the same input resolution to 3D models, feature-space reconstruction is much more effective than RGB reconstruction*.
>
>
> | Method   | Reconstruction space                                | PSNR &uarr; | SSIM &uarr; | LPIPS &darr; |
> |----------|------------------|-------------|-------------|--------------|
> | *3DGS (512x512)* | *image-space*                                 | *26.17*     | *0.778*     | *0.091*      |
> | 3DGS (64x64) | image-space           | 14.03	| 0.352	| 0.541   |
> | Image VAE reconstruction (no 3D reconstruction) | feature-space (upper-bound)               | 24.59      | 0.762       | 0.126        |
> | Ours           | feature-space              | **22.45**       | **0.667**       | **0.197**        |
>
> **Computation efficiency**: To demonstrate the efficiency advantage of feature-space 3D reconstruction, we estimated the GPU usage of 3DGS for a large outdoor scene comprising 1,100 images (1408 × 1024, captured by a drone) over 500m × 500m area. The standard 3DGS requires *40 GB* GPU memory, while our method needs only *13.1 GB*. Training standard 3DGS on such scenes is impractical with consumer-grade GPUs like the RTX 4090 of 24GB memory.
>
> **Additional pre-training overhead**: The 3D-aware autoencoder is generalizable, such that it can be directly loaded and used in real-world applications *without any per-scene fine-tuning*. Extensive out-of-domain evaluations (training on DL3DV, evaluating on NeRF-LLFF, MVImageNet, and MipNeRF-360) demonstrate the strong generalization ability of our 3D-aware autoencoder. This is different from existing latent rendering methods, such as Latent-NeRF, which require per-scene fine-tuning.

---

> > ### Comment · Reviewer_RdaN · 2024-11-27
> >
> > Thank you for the clarification regarding the motivation and positioning of this work. The response raises an interesting but concerning point about the inherent limitations of the feature-space approach.
> >
> > While the goal to "bridge the gap between 2D and 3D representations" is clear, the presented data actually reveals a fundamental limitation: the image VAE reconstruction (PSNR 24.59) represents an upper bound that is already significantly lower than direct image-space methods. This raises a critical question about the choice of working in feature space - if the very first step of encoding into feature space introduces substantial quality degradation, why choose this direction as the foundation for 3D reconstruction?

---

> ### Author Response · Authors · 2024-11-27
>
> Thank you for the beneficial discussion. **Feature-space 3D reconstruction (i.e., the Latent Radiance Field in Stage-II) only serves as one part of our entire framework.** Feature-space reconstruction evaluates the multi-view consistency of 2D encoder proposed in Stage-I, and serves as a foundation for fine-tuning the decoder in Stage-III. Again, our goal is to bridge the gap between 2D and 3D representations by constructing a 3D-aware autoencoder, rather than advancing image-space 3D reconstruction algorithms.
>
> **Why do we need 3D-consistent 2D representations?** As demonstrated by Stable Diffusion models, optimizing in the latent space instead of the image space can significantly boost generation efficiency. With a 3D-consistent latent space and photorealistic decoding capability, many tasks such as text-to-3D generation, latent NVS, sparse-view NVS, efficient NVS, and 3D latent diffusion model can be improved.

---

### Official Review · Reviewer_nTxk · 2024-11-03

**Soundness:** 3
**Presentation:** 3
**Contribution:** 2
**Rating:** 6
**Confidence:** 4

**Summary:**

The author introduces pixel-to-pixel correspondences across different viewpoints to help the VAE learn a 2D latent space with 3D awareness. Using 3D Gaussian Splatting (3DGS), they perform 3D reconstruction and rendering in the latent space to obtain 2D latent representations from specified camera poses. The rendered results are then decoded back into image space by the decoder to obtain RGB images.

Experimental results demonstrate that the resulting 2D latent space possesses a certain level of 3D perception capability and outperforms existing methods when decoding to higher-resolution images.

**Strengths:**

The author is committed to integrating 3D awareness into the 2D latent space, and the results show a significant degree of success in this endeavor. Additionally, using 3D Gaussian Splatting (3DGS) in modeling the latent space is an intriguing idea.

**Weaknesses:**

The motivation of this paper is somewhat unclear. Is the author aiming to improve reconstruction accuracy, enhance rendering speed, reduce storage space, or achieve some other application? It appears that none of these goals have been fully addressed.

**Reconstruction Accuracy**: When training the comparison methods, the author down-scaled the RGB images to the same resolution as the latent representation before training, which may be considered unfair. The VAE used by the author has been exposed to high-resolution images, while the comparison methods have not. This discrepancy could reduce the reconstruction performance of the comparison methods and impact the paper's credibility.

**Rendering Speed**: In Section 5.1, the author reports the training times for Stage 1 and Stage 3, but not for Stage 2 or for inference time. Therefore, it is challenging to conclude that the proposed method has a faster rendering or training speed compared to other methods.

**Storage Space Reduction**: In Section 5.1, the author mentions the need to "train the same number of latent 3D Gaussian splatting scenes... for Stage-III," indicating that Stage 2 is scene-specific. Compared to 3DGS, this does not seem to save much storage space.

**Other Applications**: In Section 5.3, the author suggests that their work can be used for text-to-3D generation. However, the two methods used for comparison are relatively outdated. It is recommended to compare the method with more recent approaches, such as IPDREAMER [1], to make the claim more convincing.

[1] Zeng, Bohan, et al. "Ipdreamer: Appearance-controllable 3d object generation with image prompts." arXiv preprint arXiv:2310.05375 (2023).

**Questions:**

- Could the author please provide further clarification on the motivation and applicable scenarios for this work?
- In the experimental section, it would be beneficial to employ fairer comparison methods; using low-resolution training for comparison models is not advisable.
- Please consider using more recent models to compare the text-to-3D generation capabilities of this work.

---

> ### Author Response · Authors · 2024-11-21
>
> Thank you very much for your time and effort in reviewing our work and providing the constructive comments.
>
> **Motivation.**
> This work aims to *bridge the gap between 2D and 3D representations, providing an efficient and generalizable solution for tasks requiring multi-view consistency in the feature space*. Targeting at this, there are two key challenges:
> - 1) A key challenge is that *2D latent features often lack 3D view consistency*, which impairs the performance of novel view synthesis (NVS). To address this. We introduce an efficient correspondence-aware autoencoding fine-tuning strategy that enforces multi-view consistency of 2D latent features, minimizing changes to the 2D latent space while achieving significantly better geometric consistency.
> - 2) Another key challenge is that, *applying RGB-based NVS methods to latent features would cause data distribution shift* (see below table). We mitigate this through an alignment fine-tuning process that uses latent radiance fields (LRFs) as guidance to fine-tune decoder, resulting in improved latent representation fidelity with significant PSNR gains, as shown in Table 2 and Figure 6 of the paper.
>
>
> | KL Loss (VAE encoder output)  | KL Loss (after 3DGS rendering)  |
> |------------------------------|----------------------------|
> | $3.35*10^4$                    |          $1.32*10^5$       |
>
>
> **What can our method benefit?**
> - **Input data compression of 3D reconstruction**: Our method reduces input resolutions, model storage space, and GPU usage for photorealistic NVS, which is particularly useful in cases with limited communication bandwidth and storage. For instance, some individuals may not have GPUs with large memories, where our method is an efficient solution for them to run photorealistic NVS algorithms.
>
> | Method           | Input resolution to 3D model | Training Time &darr; | GPU Usage &darr; | Storage &darr;       | Rendering FPS &uarr; | Decoding FPS &uarr; | PSNR &uarr; | SSIM &uarr; | LPIPS &darr; |
> |------------------|-------------------------|-----------------------|------------------|----------------------|----------------------|---------------------|-------------|-------------|--------------|
> | *3DGS*           | *512\*512*              | *5.9 min*             | *3 GB*           | *200.41 MB*          | *100*                | *-*                 | *26.17*     | *0.778*     | *0.091*      |
> | 3DGS/8           | 64*64                  | **3.1 min**               | 1 GB             | **59.15 MB**         | **200**              | -                   | 14.03        | 0.352       | 0.541        |
> | 3DGS-VAE         | 64*64                  | 4.8 min               | 2 GB             | 250.97 MB            | 80                   | 20                  | 20.57       | 0.595       | 0.346        |
> | Latent-NeRF      | 64*64                  | 27.2 min              | 10 GB            | 350.50 MB            | 0.09                 | 20                  | 18.16       | 0.530       | 0.432        |
> | Ours             | 64*64                  | 3.9 min           | **1 GB**         | 96.42 MB             | 180                  | 20                  | **22.45**       | **0.667**       | **0.197**        |
>
>
>
>
> - **Text-to-3D Generation**: The 3D-aware 2D representations enhance both latent and image-space text-to-3D generation frameworks. Please refer to Sec. 5.3 of the paper for this improvement on three different generation frameworks.
>
> - **Sparse View Reconstruction**: We find that the improved 3D feature consistency enables better reconstruction of missing information for few-shot NVS task. The results are even better than the state-of-the-art 3DGS methods with standard input image resolutions. The experiment is conducted on the LLFF dataset with three input views.
>
> | Method (LLFF, 3 views)                                     | PSNR &uarr; | SSIM &uarr; | LPIPS &darr; |
> |-------------------------------------------|-------------|-------------|--------------|
> | 3DGS-VAE                                  | 13.06       | 0.283       | 0.5708       |
> | 3DGS (standard resolution)                | 13.79       | 0.331       | 0.468        |
> | Mip-Splatting (standard resolution)       | 13.70       | 0.315       | 0.486        |
> | Ours                                      | **15.51**   | **0.379**   | **0.465**    |
>
>
> **What can our method potentially benefit in the future?**
> - **3D latent diffusion model**: The efficient latent radiance field would serve as a robust foundation for optimizing 3D latent diffusion models.
> - **Integration with other NVS compression methods**: Our framework could seamlessly integrate with existing NVS compression methods to enhance their efficiency and scalability.

---

> ### Author Response · Authors · 2024-11-21
>
> **Reconstruction accuracy**:
>
> The purpose of demonstrating these comparisons is to evaluate the performance of 3DGS models under the same input resolutions. When the latents share the same resolution as the RGB images, the 3D latent representation outperforms all existing methods of 3D RGB representation. This finding provides valuable insights for downstream tasks requiring latent representations while addressing limited computational constraint.
>
> **Rendering speed**:
>
>
> Please refer to the following figure for the inference speed.
>
> | Method           | Rendering FPS &uarr; | Decoding FPS &uarr; | PSNR &uarr; | SSIM &uarr; | LPIPS &darr; |
> |------------------|----------------------|---------------------|-------------|-------------|--------------|
> | 3DGS/8           | **200**             | -                   | 14.03       | 0.352       | 0.541        |
> | 3DGS-VAE         | 80                  | 20                  | 20.57       | 0.595       | 0.346        |
> | Latent-NeRF      | 0.09                | 20                  | 18.16       | 0.530       | 0.432        |
> | Ours             | 180                 | 20                  | **22.45**   | **0.667**   | **0.197**    |
>
> **Storage space reduction**:
>
> Please refer to the below table for storage space comparisons. Our method demonstrates a significant advantage in storage efficiency. While 3DGS with the same input resolution achieves slightly smaller storage, its PSNR (14.03) is much worse than our method (22.45).
> | Method           | Input Image Resolution | Storage &uarr;       | PSNR &uarr; | SSIM &uarr; | LPIPS &darr; |
> |------------------|-------------------------|----------------------|-------------|-------------|--------------|
> | *3DGS*           | *512\*512*             | *200.41 MB*          | *26.17*     | *0.778*     | *0.009*      |
> | 3DGS/8           | 64*64                  | **59.15 MB**         | 14.03       | 0.352       | 0.541        |
> | 3DGS-VAE         | 64*64                  | 250.97 MB            | 20.57       | 0.595       | 0.346        |
> | Latent-NeRF      | 64*64                  | 350.50 MB            | 18.16       | 0.530       | 0.432        |
> | Ours             | 64*64                  | 96.42 MB             | **22.45**   | **0.667**   | **0.197**    |
>
>
> **Other applications**:
>
> Please refer to Sec. 5.3 of the revised paper, where we additionally compare with the state-of-art text-to-3D generation method, GSGEN [r1]. Our method still works well on more complicated text prompts and generated geometries. For instance, in the results of vanilla GSGEN, the train on the cake appears distorted and lacks multi-view consistency. However, with the help of our model, we achieve consistent generation across views.
>
>
> [r1] Z. Chen, F. Wang, and H. Liu, “Text-to-3d using gaussian splatting”, arXiv preprint arXiv:2309.16585, 2023.

---

> > ### Comment · Reviewer_nTxk · 2024-11-24
> >
> > Thanks for your detailed feedback! My concerns are addressed and I decide to raise my rating to 6.

---

> > > ### Author Response · Authors · 2024-11-24
> > >
> > > Thank you again for your time and effort in reviewing our work and providing the constructive comments.

---

### Official Review · Reviewer_duK5 · 2024-11-03

**Soundness:** 4
**Presentation:** 3
**Contribution:** 3
**Rating:** 6
**Confidence:** 4

**Summary:**

This submission tries to resolve the problem of 3D reconstruction in the latent space via a 3-stage idea. The first stage focuses on improving the 3D awareness of the VAE's encoder via a correspondence-aware constraint on the latent space; the second stage builds a latent radiance field (LRF) to represent 3D scenes from the 3D-aware 2D representations; the last stage further introduces a VAE-Radiance Field (VAE-RF) alignment method to boost the reconstruction performance. The results generated by this pipeline out-perform the results generated by many state-of-the-arts.

**Strengths:**

There are many innovations in this work, but I think the best part is the introduction of the 3d awareness into the 2D representation training. In this part, especially the correspondence aware autoencoding is the key to the success of this overall idea.

**Weaknesses:**

There are still some weaknesses prevented me from giving a higher score, especially, the details of how to compute each component of the pipeline. Please see my questions below. In addition, some related references are missing.

**Questions:**

I'm willing to raise my score if questions below is answered:

1. How is \lambda_{ij} computed in equation (6), section 4.1? Basically, how to compute the average pose error and how it contributes to the weight \lambda_{ij}?
2. Since equation (6) becomes multi-objective optimization, does this change largely increase the training convergence time? Did you experience any convergence issues?
3. How is the inference speed of this pipeline?
4. The idea is kind of similar to CaesarNeRF: Calibrated Semantic Representation for Few-shot Generalizable Neural Rendering: https://haidongz-usc.github.io/project/caesarnerf, which also uses calibrated image feature in each 2d latent, could you cite and compare?

---

> ### Author Response · Authors · 2024-11-21
>
> Thank you very much for your time and effort in reviewing our work and providing the constructive comments.
>
> **Q1. Math definition of average pose error:**
>
> To compute $\lambda_{ij}$, we first calculate the Absolute Pose Error (APE) for each pose pair using the formula:$E_{ij} = P_i^{-1} P_j,$where $P_i$ and $P_j$ are the different camera poses respectively. After obtaining $E_{ij}$, the APE is calculated as:
>
> $$APE_{ij} = \|E_{ij} - I_{4 \times 4}\|_F,$$
>
> where $I_{4 \times 4}$ is the identity matrix and $F$ represents the Frobenius norm. In each iteration, the APE values are normalized across all image pairs to derive the weights $\lambda_{ij}$, as:
>
> $$\lambda_{ij} = \frac{APE_{ij}}{\sum_{k} {{APE}_{k}}},$$
>
> where $k$ represents each image pair within one iterations. This normalization ensures they reflect the relative contributions of each pose error in a consistent manner. This method is implemented based on the APE computation approach in the evo library [r1].
>
> Thank you for your suggestion, we will add the above explanations in the camera ready verison.
>
> [r1] Michael Grupp. evo.  https://github.com/MichaelGrupp/evo/blob/d71da47342082626b7c90404e96363e89a05cc22/notebooks/metrics.py_API_Documentation.ipynb#L191
>
> **Q2. Since equation (6) becomes multi-objective optimization, does this change largely increase the training convergence time? Did you experience any convergence issues?**
>
> The convergence time does not increase significantly. Please refer to the following table for the rendering performance corresponding to different VAE training steps. The training converges in around 50k steps. Compared with the official training steps of 250k in Stable Diffusion VAE [r2], our method only needs 1/5 training time, which is comparably efficient.
>
> | Step  | PSNR (dB)  |
> |-------|------------|
> | 50K   | 21.16      |
> | 70K  | 21.20     |
> | 100K  | 21.03      |
> | 250K  | 20.96      |
>
>
> [r2] https://huggingface.co/stabilityai/sd-vae-ft-mse
>
> **Q.3 Inference speed**:
>
> Please refer to the below table for the inference speed. Our method outperforms other latent-space approaches with the fastest inference speed and better rendering quality. Compared to image-space methods which avoid using a decoder, their image quality are significantly worse.
>
> | Method           | Rendering FPS (3DGS) &uarr; | Decoding FPS (VAE decoder) &uarr; | PSNR &uarr; | SSIM &uarr; | LPIPS &darr; |
> |------------------|----------------------|---------------------|-------------|-------------|--------------|
> | 3DGS/8           | **200**             | -                   | 14.03       | 0.352       | 0.541        |
> | 3DGS-VAE         | 80                  | 20                  | 20.57       | 0.595       | 0.346        |
> | Latent-NeRF      | 0.09                | 20                  | 18.16       | 0.530       | 0.432        |
> | Ours             | 180                 | 20                  | **22.45**   | **0.667**   | **0.197**    |
>
>
> **Q.4 Discuss CaesarNeRF**:
>
> Thank you for the helpful reference. Both CaesarNeRF and this work observe that the inconsistent feature representation would degrade NVS performance. However, our approaches are totally different. In CeasarNeRF, the inconsistent features from the reference views are calibrated by the relative camera rotations to achieve more consistent rendering results at the novel views. They employ a global semantic calibration for the entire scene, whereas we use a 3D-aware autoencoder for each image patch (corresponding to the receptive field of one pixel). The global semantic calibration fails to ensure local feature consistency, which is essential for capturing fine-grained geometric details. We did not experimentally compare with CaesarNeRF because it focuses on few-shot, feed-forward generalizable methods, while our approach is optimization-based, making the comparison unfair. We will discuss CaesarNeRF in the Related Work.

---

> > ### Comment · Reviewer_duK5 · 2024-11-24
> >
> > Thank you for your detailed reply. I will my score also after reading your answers to other reviewers.

---

> > > ### Author Response · Authors · 2024-11-27
> > >
> > > Thank you again for your time and effort in reviewing our work and providing the constructive comments.

---

### Author Response · Authors · 2024-11-21

We sincerely thank all reviewers for their time, effort, and valuable feedback, which have greatly helped us to improve this work. We would like to take this opportunity to clarify the motivations of this work, and three major experiments added during rebuttal period.

**Motivation:**
This work aims to *bridge the gap between 2D and 3D representations, providing an efficient and generalizable solution for tasks requiring multi-view consistency in the feature space*. Targeting at this, there are two key challenges:
- A key challenge is that *2D latent features often lack 3D view consistency*, which impairs the performance of novel view synthesis (NVS). To address this. We introduce a correspondence-aware autoencoding fine-tuning strategy that enforces multi-view consistency in 2D latent features. This strategy uses a correspondence loss to align 2D latent representations with RGB images by ensuring shared corresponding points, minimizing modifications to the 2D latent space while achieving geometric consistency.
- Another key challenge is that, *applying RGB-based NVS method to latent features would cause data distribution shift* (see below table). We mitigate this through an alignment fine-tuning process that uses latent radiance fields (LRFs) to fine-tune decoder. This process corrects distribution shifts by training on paired datasets, leading to improved latent representation fidelity and significant PSNR gains, as shown in Table 2 and Figure 6 of the paper.


| KL Loss (Before 3D reconstruction)  | KL Loss (After 3D reconstruction)  |
|------------------------------|----------------------------|
| $3.35*10^4$                    |          $1.32*10^5$       |

 **A comprehensive comparison of running efficiency and performance**: Our method reduces input resolutions, model storage space, and GPU usage for photorealistic NVS, which is particularly useful in cases with limited communication bandwidth and storage. For instance, some individuals may not have GPUs with large memories, where our method is an efficient solution for them to run photorealistic NVS algorithms.

| Method           | Input resolution to 3D model | Training Time &darr; | GPU Usage &darr; | Storage &darr;       | Rendering FPS &uarr; | Decoding FPS &uarr; | PSNR &uarr; | SSIM &uarr; | LPIPS &darr; |
|------------------|-------------------------|-----------------------|------------------|----------------------|----------------------|---------------------|-------------|-------------|--------------|
| *3DGS*           | *512\*512*              | *5.9 min*             | *3 GB*           | *200.41 MB*          | *100*                | *-*                 | *26.17*     | *0.778*     | *0.091*      |
| 3DGS/8           | 64*64                  | **3.1 min**               | 1 GB             | **59.15 MB**         | **200**              | -                   | 14.03        | 0.352       | 0.541        |
| 3DGS-VAE         | 64*64                  | 4.8 min               | 2 GB             | 250.97 MB            | 80                   | 20                  | 20.57       | 0.595       | 0.346        |
| Latent-NeRF      | 64*64                  | 27.2 min              | 10 GB            | 350.50 MB            | 0.09                 | 20                  | 18.16       | 0.530       | 0.432        |
| Ours             | 64*64                  | 3.9 min           | **1 GB**         | 96.42 MB             | 180                  | 20                  | **22.45**       | **0.667**       | **0.197**        |



**Sparse view reconstruction**: Thanks to reviewer Cwtt's insightful comment, we are excited to find that the improved 3D feature consistency enables better reconstruction of missing information for few-shot NVS task. Our method is even better than the state-of-the-art 3DGS methods with standard input image resolutions. The results are shown in the below table. The experiment is conducted on the LLFF dataset with 3 input views.

| Method (LLFF, 3 views)                                      | PSNR &uarr; | SSIM &uarr; | LPIPS &darr; |
|-------------------------------------------|-------------|-------------|--------------|
| 3DGS-VAE                                  | 13.06       | 0.283       | 0.5708       |
| 3DGS (standard resolution)                | 13.79       | 0.331       | 0.468        |
| Mip-Splatting (standard resolution)       | 13.70       | 0.315       | 0.486        |
| Ours                                      | **15.51**   | **0.379**   | **0.465**    |

**More recent text-to-3D generation**: We have conducted comparison with more recent text-to-3D generation method, as shown in Sec. 5.3 of the revised paper. As shown in Fig. 5, our method can boost the performance under extremely complicated text prompts, and achieve complex geometry while preserving the multi-view consistency. Our method achieves consistent generation across views.

 Please refer to our comments for each reviewer for more detailed responses.

---

### Meta-Review · Area_Chair_rgMw · 2024-12-22

**Metareview:**

The paper introduces an innovative approach that integrates 3D awareness into 2D latent space representations, specifically through a correspondence-aware autoencoding mechanism. This integration is aimed at enhancing the 3D consistency and the natural interaction between 2D latent and 3D spaces. The authors then build LRF in latent space and decode it to images.

***Strengths:***
- The method's integration of 3D awareness into 2D latent space, which is viewed as innovation enhancing the model's performance.
- Effective adaptation of various existing techniques such as 3DGS and correspondence constraints to improve 3D consistency in latent space.
- Testing across multiple datasets demonstrates the generalization capability and potential usage in different scenarios including novel view synthesis and text-to-3D generation.

***Weaknesses:***
- The paper's motivation and specific goals (such as improvement in reconstruction accuracy, rendering speed, or storage space reduction) are not clearly defined or convincingly addressed.
- Some reviewers noted the lack of detail in various components of the pipeline.
- Questions about the computational advantages or the justification for added complexity in operating within latent space

The authors have provided detailed rebuttal with further experiments and explanation. After careful consideration and discussion, we are pleased to inform this paper is accepted. The decision to accept the paper is from following considerations:
- Novel approach to incorporating 3D awareness into 2D latent space
- Despite some limitations, the paper provides expriments on multiple datasets, showing better generalization and potential for practical applications.
- The authors effectively addressed many of the initial concerns raised during the rebuttal period.

Overall, while the paper has some areas that could benefit from further refinement, its strengths in innovation, and empirical validation are sufficient to merit acceptance. The paper provides a way to think about and integrate 3D data within 2D latent frameworks and we think there might be potential for future direction. The authors are suggeted to carefully polish the paper, from motivation presentation to advantage demonstration, based on the reviewers' comments.

**Additional Comments On Reviewer Discussion:**

There is a debate focusing on the quality of novel view synthesis. Reviewer RdaN expressed continued reservations about the paper, particularly highlighting significant quality degradation in NV rendering compared to image-space methods. This raises questions about the effectiveness of the feature-space approach. In contrast, Reviewer duK5 voiced support for the authors, optimistic that the current limitations in image VAE reconstruction will not be a long-term issue, and advocated for the potential of this work.

The AC placed greater emphasis on the potential of the research rather than its current limitations. The AC suggested that the authors reorganize the paper to specifically highlight the advantages of using latent space, rather than making broad claims where the actual   benefits are just limited. For example, in the rebuttal, the authors added experiments on sparse view reconstruction and demonstrated advantages in few-shot cases. This new result was also one of the factor in the overall decision to recommend acceptance.

---

### Decision · Program_Chairs · 2025-01-22

Accept (Poster)